# A Prefrontal Cortex-inspired Architecture for Planning in Large Language Models

## Abstract

Large language models (LLMs) demonstrate impressive performance on a wide variety of tasks, but they often struggle with tasks that require multi-step reasoning or goal-directed planning. To address this, we take inspiration from the human brain, in which planning is accomplished via the recurrent interaction of specialized modules in the prefrontal cortex (PFC). These modules perform functions such as conflict monitoring, state prediction, state evaluation, task decomposition, and task coordination. We find that LLMs are sometimes capable of carrying out these functions in isolation, but struggle to autonomously coordinate them in the service of a goal. Therefore, we propose a black box architecture with multiple LLM-based (GPT-4) modules. The architecture improves planning through the interaction of specialized PFC-inspired modules that break down a larger problem into multiple brief automated calls to the LLM. We evaluate the combined architecture on three challenging planning tasks – graph traversal, Tower of Hanoi, and logistics – finding that it yields significant improvements over standard LLM methods (e.g., zero-shot prompting, in-context learning, and chain-of-thought). These results demonstrate the benefit of utilizing knowledge from cognitive neuroscience to improve planning in LLMs.

## 1 Introduction

Large Language Models (LLMs) (Devlin et al., 2019; Brown et al., 2020) have recently emerged as highly capable generalist systems with a surprising range of emergent capacities (Srivastava et al., 2022; Wei et al., 2022a; Webb et al., 2023). They have also sparked broad controversy, with some suggesting that they are approaching general intelligence (Bubeck et al., 2023), and others noting a number of significant deficiencies (Mahowald et al., 2023). A particularly notable shortcoming is their poor ability to plan or perform faithful multi-step reasoning (Valmeekam et al., 2023; Dziri et al., 2023). Recent work (Momennejad et al., 2023) has evaluated the extent to which LLMs might possess an emergent capacity for planning and exploiting *cognitive maps*, the relational structures that humans and other animals utilize to perform planning (Tolman, 1948; Tavares et al., 2015; Behrens et al., 2018). This work found that a variety of LLMs, ranging from small, open-source models (e.g., LLaMA-13B and Alpaca-7B) to large, state-of-the-art models (e.g., GPT-4), displayed systematic shortcomings in planning tasks that suggested an inability to reason about cognitive maps. Common failure modes included a tendency to 'hallucinate' (e.g., to imagine non-existent paths), and to fall into loops. This work raises the question of how LLMs might be improved so as to enable a capacity for planning.

In the present work, we take a step toward improving planning in LLMs, by taking inspiration from the planning mechanisms employed by the human brain. Planning is generally thought to depend on the prefrontal cortex (PFC) (Owen, 1997; Russin et al., 2020; Brunec & Momennejad, 2022; Momennejad et al., 2018; Momennejad, 2020; Mattar & Lengyel, 2022), a region in the frontal lobe that is broadly involved in executive function, decision-making, and reasoning (Miller & Cohen, 2001). Research in cognitive neuroscience has revealed the presence of several subregions or modules within the PFC that appear to be specialized to perform certain functions. These include functions such as conflict monitoring (Botvinick et al., 1999); state prediction and state evaluation (Wallis, 2007; Schuck et al., 2016); and task decomposition and task coordination (Ramnani & Owen, 2004; Momennejad & Haynes, 2012; 2013). Human planning then emerges through the

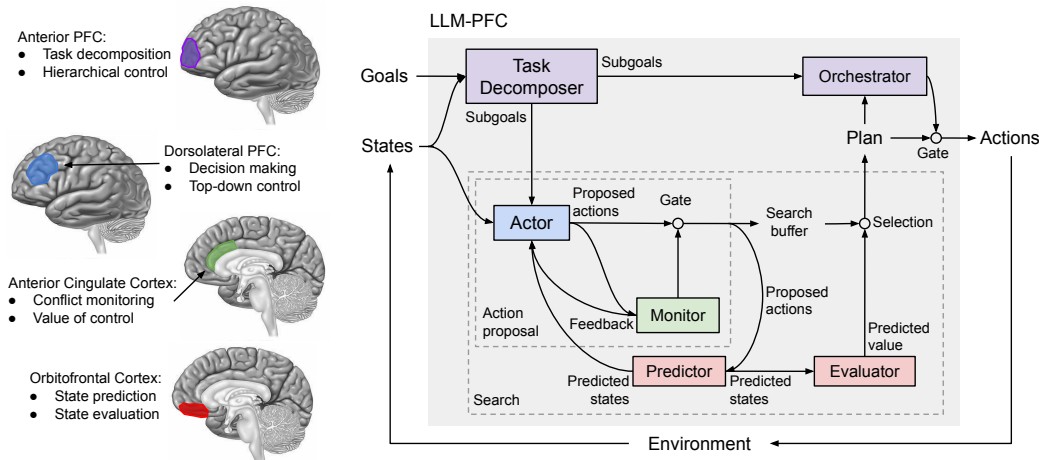

Figure 1: **LLM-PFC architecture.** The agent receives states from the environment and high-level goals. These are processed by a set of specialized LLM modules. The TaskDecomposer receives high-level goals and generates a series of subgoals. The Actor generates proposed actions given a state and a subgoal. The Monitor gates these proposed actions based on whether they violate certain constraints (e.g., task rules) and provides feedback to the Actor. The Predictor predicts the next state given the current state and a proposed action. The Evaluator is used to estimate the value of a predicted state. The Predictor and Evaluator are used together to perform tree search. The Orchestrator determines when each subgoal has been achieved, and when the final goal has been achieved, at which point the plan is emitted to the environment as a series of actions. These modules are inspired by the specific PFC subregions depicted on the left.

coordinated and recurrent interactions among these specialized PFC modules, rather than through the activity of a single, monolithic system.

An interesting observation is that LLMs often seem to display some of these capacities when probed in isolation, even though they are unable to reliably integrate and deploy these capacities in the service of a goal. For instance, Momennejad et al. (2023) noted that LLMs often attempt to traverse invalid or hallucinated paths in planning problems (e.g., to move between rooms that are not connected), even though they can correctly identify these paths as invalid when probed separately. This suggests the possibility of a PFC-inspired approach, in which planning is carried out through the coordinated activity of multiple LLM modules, each of which is specialized to perform a distinct process.

With this goal in mind, we propose LLM-PFC (Figure 1), an architecture composed of modules that are specialized to perform specific PFC-inspired functions. Each module consists of an LLM instance (GPT-4), constructed through a combination of prompting and few-shot in-context learning. We specifically propose modules that perform error monitoring, action proposal, state prediction, state evaluation, task decomposition, and task coordination. It is suggested that the coordinated activity of multiple PFC subregions performs tree search during planning (Owen, 1997; Daw et al., 2005; Wunderlich et al., 2012; Doll et al., 2015). Thus, our approach combines action proposal, state prediction, and state evaluation to perform tree search.

We evaluate LLM-PFC on three challenging planning tasks. First, we performed controlled experiments on a set of graph traversal tasks using the CogEval protocol (Momennejad et al., 2023). These tasks require navigation in novel environments based on natural language descriptions, and have been shown to be extremely challenging for LLMs, including GPT-4. Second, we investigate Tower of Hanoi (ToH), a classic problem solving task that requires multi-step planning (Simon, 1975), and for which performance is known to be heavily dependent on PFC function (Goel & Grafman, 1995; Fincham et al., 2002). Finally, we investigate a more complex, real-world planning task involving logistics (transportation of goods) (Valmeekam et al., 2023). We find that our approach significantly improves LLM performance on all three tasks. Ablation experiments further indicate

that each of the individual modules plays an important role in the overall architecture's performance. Taken together, these results indicate the potential of a PFC-inspired approach to improve the reasoning and planning capabilities of LLMs.

## 2 Approach

The LLM-PFC architecture is constructed from a set of a specialized LLM modules, each of which performs a specific PFC-inspired function. In the following sections, we first describe the functions performed by each module, and then describe how they interact to generate a plan.

### 2.1 Modules

LLM-PFC contains the following specialized modules, each constructed from a separate LLM instance through a combination of prompting and few-shot ($\leq 3$ examples) in-context learning (described in greater detail in section A.6):

• **TaskDecomposer**. The TaskDecomposer receives the current state $x$ and a goal $y$ and generates a set of subgoals $Z$ that will allow the agent to gradually work toward its final goal. This module is inspired by the anterior PFC (aPFC), which is known to play a key role in task decomposition through the generation and maintenance of subgoals (Ramnani & Owen, 2004). In the present work, the TaskDecomposer is only utilized to generate a single intermediate goal, though in future work we envision that it will be useful to generate a series of multiple subgoals.

• **Actor**. The Actor receives the current state $x$ and a subgoal $z$ and proposes $B$ potential actions $A = a_{b=1} \ldots a_{b=B}$. The Actor can also receive feedback $\epsilon$ from the Monitor about its proposed actions. This module can be viewed as being analogous to the dorsolateral PFC (dlPFC) which plays a role in decision making through top-down control and guidance of lower-order premotor and motor regions (Miller & Cohen, 2001).

• **Monitor**. The Monitor assesses the actions proposed by the Actor to determine whether they are valid (e.g., whether they violate the rules of a task). It emits an assessment of validity $\sigma$, and also feedback $\epsilon$ in the event the action is deemed invalid. This module is inspired by the Anterior Cingulate Cortex (ACC), which is known to play a role in conflict monitoring (Botvinick et al., 1999), i.e., detecting errors or instances of ambiguity.

• **Predictor**. The Predictor receives the current state $x$ and a proposed action $a$ and predicts the resulting next state $\tilde{x}$. The Predictor is inspired by the Orbitofrontal cortex (OFC), which plays a role in estimating and predicting task states. In particular, it has been proposed that the OFC plays a key role in encoding cognitive maps: representations of task-relevant states and their relationships to one another (Schuck et al., 2016).

• **Evaluator**. The Evaluator receives a next-state prediction $\tilde{x}$ and produces an estimate of its value $v$ in the context of goal $y$. This is accomplished by prompting the Evaluator (and demonstrating via a few in-context examples) to estimate the minimum number of steps required to reach the goal (or subgoal) from the current state. The Evaluator is also inspired by the OFC which, in addition to predicting task states, plays a key role in estimating the motivational value of those states Wallis (2007).

• **Orchestrator**. The Orchestrator receives the current state $x$ and a subgoal $z$ and emits an assessment $\Omega$ of whether the subgoal has been achieved. When the Orchestrator determines that all subgoals (including the final goal) have been achieved, the plan is emitted to the environment as a series of actions. This module is also inspired by the aPFC, which is thought to both identify subgoals and coordinate their sequential execution (Ramnani & Owen, 2004).

### 2.2 Action proposal loop

The Actor and Monitor interact via the ProposeAction function (Algorithm 1). The Actor proposes actions which are then gated by the Monitor. If the Monitor determines that the actions are invalid (e.g., they violate the rules of a task), feedback is provided to the Actor, which then proposes an alternative action. In the brain, a similar process is carried out by interactions between the ACC and dorsolateral PFC (dlPFC). The ACC is thought to recruit the dlPFC under conditions of conflict

(e.g., errors or ambiguity), which then acts to resolve the conflict through top-down projections to lower-order control structures (e.g., premotor and motor cortices) (Miller & Cohen, 2001; Shenhav et al., 2013).

---

**Algorithm 1: Action proposal loop.** ProposeAction takes a state $x$ and a goal $y$ and generates $B$ potential actions $A = a_{b=1} \ldots a_{b=B}$. This is implemented via a loop, in which the Actor first proposes potential actions, and the Monitor then assesses those actions according to certain constraints (e.g., task rules), providing feedback if any of the actions are deemed to be invalid. This continues until the proposed actions are considered valid. See Sections A.6.2 and A.6.3 for more details.

---

**Function** ProposeAction $(x, y, B)$ **:**

    $\sigma \leftarrow$ false             // Initialize validity
    $E \leftarrow \{\}$            // Initialize feedback
    **while** $\sigma$ is false **do**
        $A \leftarrow \text{Actor}(x, y, E, B)$      // Sample B actions
        $\sigma, \epsilon \leftarrow \text{Monitor}(x, A)$    // Determine validity and provide feedback
        $E \leftarrow E \cup \{\epsilon\}$      // Accumulate feedback
    **end**
**return** $A$

---

## 2.3 SEARCH LOOP

ProposeAction is further embedded in a Search loop (Algorithm 2). The actions emitted by ProposeAction are passed to the Predictor, which predicts the states that will result from these actions. A limited tree search is then performed, starting from the current state, and then exploring $B$ branches recursively to a depth of $L$ layers. Values are assigned to the terminal states of this search by the Evaluator, and the action leading to the most valuable predicted state is selected. This approach mirrors that of the human brain, in which search is thought to be carried out through the coordinated activity of multiple regions within the PFC, including dlPFC, ACC, and OFC (Owen, 1997; Mattar & Lengyel, 2022).

---

**Algorithm 2: Search loop.** Tree search with a depth of $L$ layers, with $B$ branches at each layer $l$. For each branch, a proposed action is sampled, and the Predictor predicts the next state $\tilde{x}$. This process continues recursively until the terminal layer $L$, at which point the value $v_{l=L}$ of the terminal states is estimated by the Evaluator. The values are backpropagated to their parent states in the first layer, and the action that leads to the most valuable state is selected. In our implementation, we accelerate this process by caching the actions and predicted states from deeper search layers and then reusing them in subsequent searches. We also employ the Orchestrator to prematurely terminate search if the goal state is achieved.

---

**Function** Search $(l, L, B, x, y)$ **:**

    $V_l \leftarrow \{\}$           // Initialize value record
    $\tilde{X}_l \leftarrow \{\}$           // Initialize next-state record
    $A_l \leftarrow \text{ProposeAction}(x, y, B)$       // Propose B actions
    **for** $b$ in $1 \ldots B$ **do**
        $\tilde{x}_{lb} \leftarrow \text{Predictor}(x, A_{lb})$      // Predict next state
        $\tilde{X}_l \leftarrow \tilde{X}_l \cup \{\tilde{x}_{lb}\}$      // Update next-state record
        $\Omega \leftarrow \text{Orchestrator}(\tilde{x}_{lb}, y)$    // Terminate search if goal achieved
        **if** $l < L$ and $\Omega$ is false **then**
            $a_{l+1}, \tilde{x}_{l+1}, v_{l+1} \leftarrow \text{Search}(l+1, L, B, \tilde{x}_{lb}, y)$    // Advance search depth
            $V_l \leftarrow V_l \cup \{v_{l+1}\}$      // Update value record
        **else**
            $v_{lb} \leftarrow \text{Evaluator}(\tilde{x}_{lb}, y)$      // Evaluate predicted state
            $V_l \leftarrow V_l \cup \{v_{lb}\}$      // Update value record
        **end**
    **end**
    $v_l \leftarrow \max(V_l)$      // Maximum value (randomly sample if equal value)
    $a_l \leftarrow A_{l\,\text{argmax}(V_l)}$      // Select action
    $\tilde{x}_l \leftarrow \tilde{X}_{l\,\text{argmax}(V_l)}$      // Predicted next-state
**return** $a_l, \tilde{x}_l, v_l$

---

---

**Algorithm 3: LLM-PFC.** LLM-PFC takes a state $x$ and a goal $y$ and generates a plan $P$, a series of actions with a maximum length of $T$. The TaskDecomposer first generates a set of subgoals $Z$. The agent then pursues each individual subgoal $z$ in sequence, followed by the final goal $y$. At each time step, Search is called to generate an action and a predicted next-state. Actions are added to the plan until the Orchestrator determines that the goal has been achieved, or the plan reaches the maximum length $T$.

---

**Function** LLM-PFC $(x, y, T, L, B)$ **:**
 $P \leftarrow []$                  // Initialize plan
 $Z \leftarrow \text{TaskDecomposer}(x, y)$       // Generate subgoals
 **for** $g$ in $1 \ldots \text{length}(Z) + 1$ **do**
  **if** $g \leq \text{length}(Z)$ **then**
   $z \leftarrow Z_g$          // Update current subgoal
  **else**
   $z \leftarrow y$           // Final goal
  **end**
  $\Omega \leftarrow \text{Orchestrator}(x, z)$    // Initialize subgoal assessment
  **while** $\Omega$ is false and $\text{length}(P) < T$ **do**
   $a, x, v \leftarrow \text{Search}(l = 1, L, B, x, z)$     // Perform search
   $P \leftarrow [P, a]$          // Update plan
   $\Omega \leftarrow \text{Orchestrator}(x, z)$   // Determine if subgoal is achieved
  **end**
 **end**
**return** $P$

---

## 2.4 PLAN GENERATION

Algorithm 3 describes the complete LLM-PFC algorithm. To generate a plan, a set of subgoals is first generated by the TaskDecomposer based on the final goal and current state. These subgoals are then pursued one at a time, utilizing the Search loop to generate actions until the Orchestrator determines that the subgoal has been achieved. The actions are accumulated in a plan buffer $P$ until either the Orchestrator determines that the final goal has been reached, or the maximum allowable number of actions $T$ are accumulated. This approach is inspired by the role that aPFC plays in task decomposition. This involves the decomposition of tasks into smaller, more manageable tasks, and the coordinated sequential execution of these component tasks (Ramnani & Owen, 2004).

## 3 EXPERIMENTS

### 3.1 TASKS

**Graph Traversal.** We performed controlled experiments on four multi-step planning tasks based on graph traversal using the CogEval protocol (Momennejad et al., 2023). Natural language descriptions of a graph are provided with each node assigned to a room (e.g., 'room 4 is connected to room 7'). We focused on a particular type of graph (Figure 4) with community structure (Schapiro et al., 2013) previously found to be challenging for a wide variety of LLMs. The first task, Valuepath, involves finding the shortest path from a given room that results in the largest reward possible. A smaller reward and a larger reward are located at two different positions in the graph. We fixed the two reward locations, and created 13 problems based on different starting locations. The second task, Steppath, involves finding the shortest path between a pair of nodes. We evaluated problems with an optimal shortest path of 2, 3, or 4 steps. We generated 20 problems for each of these conditions by sampling different starting and target locations.

The other two tasks, Detour and Reward Revaluation, involve modifications to the Valuepath task that test for flexibility in planning. In these tasks, the problem description and in-context examples for the Valuepath task are presented, and a single Valuepath problem is solved as in the original task. The task is then modified in-context in one of two ways. In the Detour task, an edge is removed from the graph and replaced with a new edge (e.g., 'the door from room 1 to room 11 is locked and now room 13 is connected to room 11'). In the Reward Revaluation task, the value associated with the two reward locations is changed (e.g., 'the reward of the chest in room 8 has been changed to 12 and

the reward of the chest in room 15 has been changed to 48'). As with the Valuepath task, the Detour and Reward Revaluation tasks each involved 13 problems based on different starting locations.

**Tower of Hanoi.** We also investigated a classic multi-step planning task called the Tower of Hanoi (ToH) (Figure 5). In the original formulation, there are three pegs and a set of disks of different sizes. The disks are stacked in order of decreasing size on the leftmost peg. The goal is to move all disks to the rightmost peg, such that the disks are stacked in order of decreasing size. There are a couple of rules that determine which moves are considered valid. First, a disk can only be moved if it is at the top of its stack. Second, a disk can only be moved to the top of another stack if it is smaller than the disks in that stack (or if the peg is empty). More complex versions of the task can be created by using a larger number of disks.

We designed an alternative formulation of this task in which the inputs are text-based rather than visual. In this alternative formulation, three lists (A, B, and C) are used instead of the three pegs, and a set of numbers (0, 1, 2, and so on) is used instead of disks of different sizes. The goal is to move all numbers so that they are arranged in ascending order in list C. The rules are isomorphic to ToH. First, a number can only be moved if it is at the end of a list. Second, a number can only be moved to the end of a new list if it is larger than all the numbers in that list. Note that although this novel formulation is isomorphic to ToH (and equally complex), it does not share any surface features with the original ToH puzzle (disks, pegs, etc.), and thus GPT-4 cannot rely on exposure to descriptions of ToH in its training data to solve the problem. We created multiple problem instances by varying the initial state (the initial positions of the numbers). This resulted in 26 three-disk problems and 80 four-disk problems.

**Logistics.** To assess the ability to generate plans in more real-world settings, we investigated a logistics plan generation task involving the transportation of goods between cities using airplanes and trucks (more details can be found in Valmeekam et al. (2023)).

## 3.2 BASELINES

We compared our model to several baseline methods. The first method involved asking GPT-4 (zero-shot) to provide the solution step by step. For the second method, in-context learning (ICL), we provided GPT-4 with a few in-context examples of a complete solution. We provided two examples for ToH and Valuepath, and 3 examples (one each for 2, 3, and 4 steps) for Steppath. The third method was chain-of-thought (CoT) (Wei et al., 2022b). For this method, the in-context examples were annotated with a series of intermediate computations that break down the planning process into multiple steps (see Sections A.6.7-A.6.9 for example baseline prompts).

We also evaluated tree-of-thought (ToT) (Yao et al., 2023). Similar to LLM-PFC, ToT combines multiple LLM modules – a generator and an evaluator – to perform tree search. We implemented the generator by combining the prompts from our Actor and Predictor modules, and implemented the evaluator by combining the prompts from our Monitor and Evaluator modules (see Sections A.6.10-A.6.11). We used the codebase provided by Yao et al. (2023). Plans were terminated when the predicted state matched the goal state (based on a groundtruth evaluation, as opposed to requiring the model to make this determination for itself as in LLM-PFC). For each problem, we selected the best out of five proposed plans (again based on a groundtruth evaluation). Although these decisions arguably gave ToT an advantage relative to LLM-PFC, we chose to evaluate ToT in this way so as to give it the best possible chance of performing well in our task setting.

Finally, we evaluated multi-agent debate (MAD), using the codebase from Du et al. (2023). In this approach, similar to LLM-PFC, a solution is generated through the interaction between multiple LLM instances (each instance was equivalent to the GPT-4 ICL baseline); however, unlike LLM-PFC, these instances are not specialized to perform specific functions.

## 4 RESULTS

Figure 2 shows the results on the four graph traversal tasks (see Section A.4 for all results in Table form). On the Valuepath task, LLM-PFC solved 100% of problems, significantly outperforming both baselines. On the Steppath task, LLM-PFC displayed perfect performance for 2-step and 3-step paths, and near-perfect performance for 4-step paths, again significantly outperforming both

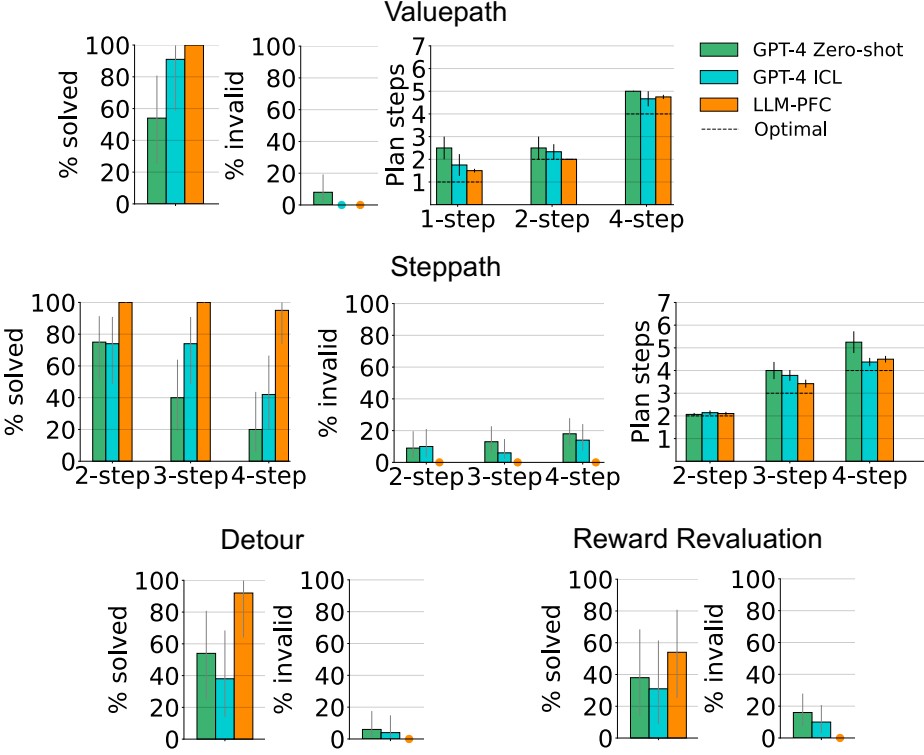

Figure 2: **Graph traversal results.** Top row: Valuepath results. Middle row: Steppath results. Bottom row: Detour and Reward Revaluation results. '% solved' indicates percentage of problems solved without proposing invalid actions (↑ better). '% invalid' indicates percentage of moves that are invalid (↓ better). 'Plan steps' indicates number of steps in plan for solved problems only (therefore excluding many problems for the baseline models; ↓ better). GPT-4 Zero-shot and ICL baselines are deterministic, and therefore a single run was performed on all problems. Note that LLM-PFC did not employ tree search on the Steppath task, and did not employ task decomposition on any of the graph traversal tasks, as the performance of the model was already at ceiling without these components. Without tree search, LLM-PFC's performance is deterministic, and therefore only a single run was performed on the Steppath task. Gray error bars reflect 95% binomial confidence intervals (for models evaluated on a single run). For Valuepath, we performed 5 runs with LLM-PFC, and present average performance ± the standard error of the mean (black error bars).

baselines. Notably, LLM-PFC's proposed plans were close to the optimal number of steps for both tasks. LLM-PFC also significantly outperformed both baselines on the Detour and Reward Revaluation tasks, with near-perfect performance in the Detour task. This demonstrates that LLM-PFC can flexibly adjust to new circumstances when generating plans. Finally, the model did not propose any invalid actions in any of the four tasks (e.g., it did not hallucinate the presence of non-existent edges), due to the filtering of invalid actions by the Monitor.

Figure 3 shows the results on Tower of Hanoi (ToH). LLM-PFC demonstrated a significant improvement both in terms of the number of problems solved (left) and the number of invalid actions proposed (right). On 3-disk problems, LLM-PFC yielded a nearly seven-fold improvement in the number of problems solved over zero-shot performance, and also significantly outperformed standard in-context learning (ICL), chain-of-thought (CoT ICL), tree-of-thought (ToT), and multi-agent debate (MAD). For the problems that LLM-PFC solved, the average plan length (5.4) was close to the optimal number of moves (4.4). The model also demonstrated some ability to generalize out-of-distribution (OOD) to more complex 4-disk problems (not observed in any in-context examples), whereas the baseline models solved close to 0% of these problems. Notably, LLM-PFC did

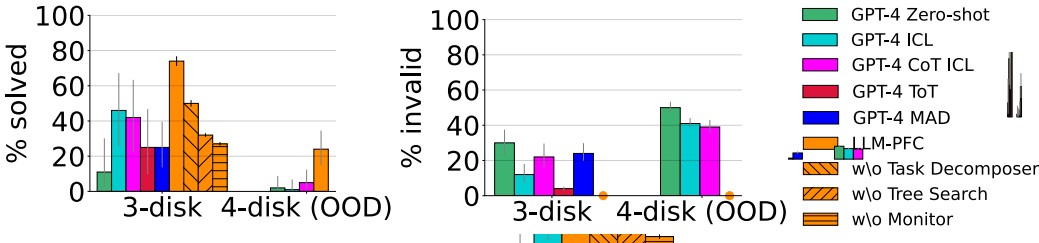

Figure 3: **Tower of Hanoi (ToH) results.** '% solved' indicates percentage of problems solved without proposing invalid actions (↑ better). '% invalid' indicates percentage of moves that are invalid (↓ better). Note that 4-disk problems are out-of-distribution (OOD). GPT-4 Zero-shot, ICL, and CoT ICL baselines are deterministic and reflect a single run. Gray error bars reflect 95% binomial confidence intervals. Dots reflect values of 0%. LLM-PFC results for 3-disk problems reflect the average over 5 runs ± the standard error of the mean (black error bars). LLM-PFC results for 4-disk problems reflect a single run, due to the high computational cost of multiple runs.

not propose any invalid actions, even on OOD 4-disk problems, whereas the baselines proposed a significant number of invalid actions.

Finally, we found that LLM-PFC also significantly improved performance on the logistics task, successfully solving **31% (62/200)** of problems[1], whereas the GPT-4 ICL baseline solved only 10.5% (21/200) of problems, and GPT-4 zero-shot solved only 7.5% (15/200) of problems (as reported in the original paper (Valmeekam et al., 2023)). This demonstrates the potential of LLM-PFC to be beneficial in more real-world planning domains.

### 4.1 ABLATION STUDY

We also carried out an ablation study to determine the relative importance of each of LLM-PFC's major components, focusing on the 3-disk ToH problems. Figure 3 (left) shows the results. We found that the MONITOR was the most important component, as ablating this module resulted in significantly fewer solved problems, due primarily to an increased tendency to propose invalid moves (31% invalid moves vs. 0% for other ablation models). Ablating the tree search and TASKDECOMPOSER module also resulted in significantly fewer solved problems. Overall, these results suggest that all major components played an important role in the model's performance.

### 5 RELATED WORK

Early work in AI formalized planning as a problem of search through a combinatorial state space, typically utilizing various heuristic methods to make this search tractable (Newell & Simon, 1956; Newell et al., 1959). Problems such as ToH figured prominently in this early research (Simon, 1975), as it affords the opportunity to explore ideas based on hierarchical or recursive planning (in which a larger problem is decomposed into a set of smaller problems). Our proposed architecture adopts some of the key ideas from this early work, including tree search and hierarchical planning.

A few recent studies have investigated planning in LLMs. These studies suggest that, although LLMs can perform relatively simple planning tasks (Huang et al., 2022), and can learn to make more complex plans given extensive domain-specific fine-tuning (Pallagani et al., 2022; Wu et al., 2023), they struggle on tasks that require zero-shot or few-shot generation of complex multi-step plans (Valmeekam et al., 2023; Momennejad et al., 2023). These results also align with studies that have found poor performance in tasks that involve other forms of extended multi-step reasoning, such as arithmetic (Dziri et al., 2023). Our approach is in large part motivated by the poor planning and reasoning performance exhibited by LLMs in these settings.

---

[1]Note that we did not use tree search or the TASKDECOMPOSER on these problems. Incorporating these components may further improve the performance of LLM-PFC on this task.

Some recent approaches have employed various forms of heuristic search to improve performance in LLMs (Lu et al., 2021; Zhang et al., 2023), but these approaches have generally involved search at the level of individual tokens. This is in contrast to our approach, in which search is performed at the more abstract level of task states (described in natural language). This is similar to other recently proposed black-box approaches in which 'thoughts' – meaningful chunks of natural language – are utilized as intermediate computations to solve more complex problems. These approaches include scratchpads (Nye et al., 2021), chain-of-thought (Wei et al., 2022b), tree-of-thoughts (Yao et al., 2023), reflexion (Shinn et al., 2023), Society of Mind (Du et al., 2023), and Describe-Explain-Plan-Select (Wang et al., 2023). All of these approaches can be viewed as implementing a form of controlled, or 'system 2', processing (as contrasted with automatic, or 'system 1', processing) (Schneider & Shiffrin, 1977; Sloman, 1996; Kahneman, 2011). In the brain, these controlled processes are strongly associated with the prefrontal cortex (Miller & Cohen, 2001). Therefore, in the present work, we leveraged knowledge from cognitive neuroscience about the modular properties of the PFC. The resulting architecture shares some components with other black box approaches (e.g., tree search (Yao et al., 2023)), but also introduces a number of new components (error monitoring, task decomposition, task coordination, state/action distinction), and combines these components in a novel manner inspired by the functional organization of the human brain (see Section A.1).

There have also been a number of proposals for incorporating modularity into deep learning systems, including neural module networks (Andreas et al., 2016), and recurrent independent mechanisms (Goyal et al., 2019). Our approach is distinguished from these approaches by the proposal of modules that perform specific high-level component processes, based on knowledge of specific subregions within the PFC. Finally, our approach is closely related to a recent proposal to augment deep learning systems with PFC-inspired mechanisms (Russin et al., 2020). LLM-PFC can be viewed as a concrete framework for accomplishing this goal.

## 6 CONCLUSION AND FUTURE DIRECTIONS

In this work, we have proposed the LLM-PFC architecture, an approach aimed at improving the planning ability of LLMs by taking inspiration from the modular architecture of the human PFC. In experiments on three challenging planning domains, we found that LLM-PFC significantly improved planning performance over standard LLM methods. While these results represent a significant step forward, there is still room for improvement. In particular, the model has less than optimal performance on Tower of Hanoi, the Reward Revaluation graph traversal task, and the Logistics planning task (Valmeekam et al., 2023) (see Section A.5). This may be due in part to the inherent limitations of prompting and in-context learning as methods for the specialization of LLM-PFC's modules. A promising avenue for further improvement may be to jointly fine-tune the modules across a range of diverse tasks (which requires open-source models), rather than relying only on black box methods (our only option with GPT-4). A white-box approach would also eliminate the need for task-specific prompts, and potentially enable zero-shot planning on novel tasks.

LLM-PFC also has important implications for neuroscientific models of PFC function. Though much work has characterized the function of individual PFC subregions, there has been less emphasis on the development of integrative models in which these functions interact to carry out coordinated behavior. The present work represents a first step in that direction. An important next step will be to directly evaluate LLM-PFC as a model of neural data, which may then lead to further refinements of the model. We look forward to investigating these possibilities in future work.

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

# A  APPENDIX

## A.1  EXTENDED RELATED WORK

In this section, we consider in more detail how LLM-PFC relates to existing black-box LLM approaches:

- Similar to LLM-PFC, both scratchpad (Nye et al., 2021) and chain-of-thought (CoT) (Wei et al., 2022b) decompose a problem into intermediate computations. However, unlike LLM-PFC, neither scratchpad nor CoT factorize these intermediate computations into specialized modules.

- Tree-of-thought (ToT) (Yao et al., 2023) introduces some degree of factorization, but the factorization is not as extensive as in LLM-PFC. The 'generator' module in ToT carries out a combination of the functions carried out by both the Actor (action proposal) and the Predictor (prediction of the states that will result from these actions) in LLM-PFC. The 'evaluator' module in ToT carries out a combination of the functions carried out by both the Monitor (error detection) and the Evaluator (prediction of state value) in LLM-PFC. ToT does not contain any component that carries out the functions of the TaskDecomposer (subgoal proposal) and the Orchestrator (autonomously determining when a goal or subgoal has been achieved).

- Multi-agent debate (i.e., Society of Mind) (Du et al., 2023) involves the interaction of multiple LLM instances; but, unlike LLM-PFC, these model instances are not specialized to perform specific functions.

- Similar to LLM-PFC, reflexion (Shinn et al., 2023) involves an element of self evaluation of proposed policies, but this depends on interaction with the external environment to determine the outcome of each policy. In LLM-PFC, this self evaluation process is entirely internal to the agent.

- Describe-Explain-Plan-Select (Wang et al., 2023) involves the coordination of multiple modules, but the approach is specific to settings involving an agent that is spatially embedded in a 2D environment. For instance, the method utilizes the spatial proximity of objects to the agent for prioritization of subgoals.

## A.2  EXPERIMENT DETAILS

We implemented each of the modules using a separate GPT-4 (32K context, '2023-03-15-preview' model index, Microsoft Azure openAI service) instance through a combination of prompting and few-shot in-context examples. We set Top-p to 0 and temperature to 0, except for the Actor (as detailed in section A.6.2). The Search loop explored $B = 2$ branches recursively for a depth $L = 2$.

For ToH, we used two randomly selected in-context examples of three-disk problems, and a description of the problem in the prompts for all the modules. For the graph traversal tasks, we used two in-context examples for all modules, except for the Actor and Evaluator in the Steppath task, where we used three in-context examples, one each for 2-, 3-, and 4-step paths. The prompt also described the specific task that was to be performed by each module (e.g., monitoring, task decomposition). For more details about the prompts and specific procedures used for each module, see Section A.6.

For three-disk problems, we allowed a maximum of $T = 10$ actions per problem, and evaluated on 24 out of 26 possible problems (leaving out the two problems that were used as in-context examples for the Actor). We also evaluated on four-disk problems, for which we allowed a maximum of $T = 20$ actions per problem. The same three-disk problems were used as in-context examples, meaning that the four-disk problems tested for out-of-distribution (OOD) generalization. For the graph traversal tasks, we allowed a maximum of $T = 6$ actions per problem.

We didn't use a separate Predictor for the graph traversal tasks, since the action proposed by the Actor gives the next state. We also did not include the TaskDecomposer for these tasks, and did not use the Search loop for the Steppath task, as the model's performance was already at ceiling without the use of these components.

A.3   ILLUSTRATION OF PLANNING TASKS

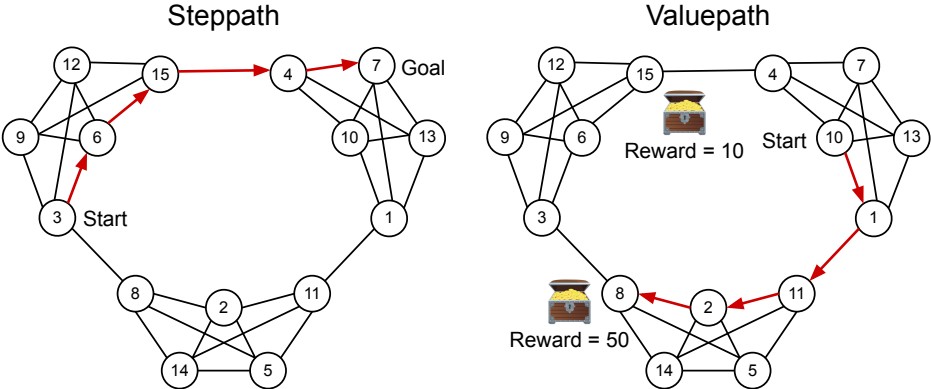

Figure 4: **Graph Traversal.** We investigated two graph traversal tasks utilizing a challenging graph with community structure. **Steppath:** Find shortest path between two nodes, e.g. node 3 and node 7. **Valuepath:** Find shortest path from starting location (e.g., node 10) to location with maximum reward (node 8 in depicted example).

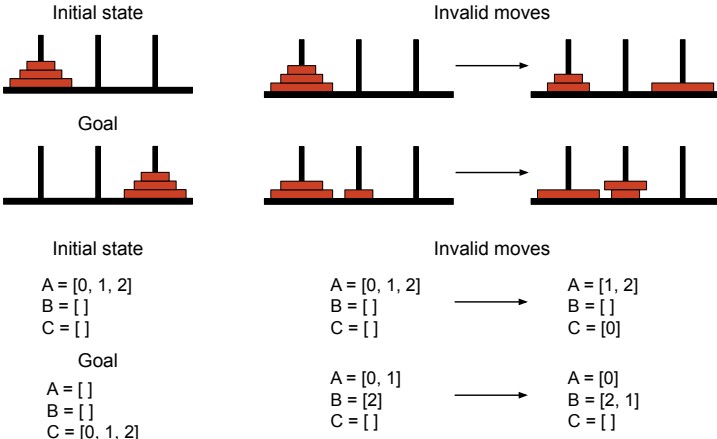

Figure 5: **Tower of Hanoi. Top:** Depiction of the Tower of Hanoi (ToH) puzzle. Disks are stacked in order of decreasing size on the leftmost peg. The goal is to move these disks so that they are stacked in order of decreasing size on the rightmost peg. Only the disk on the top of the stack may be moved, and a disk can only be placed on top of larger disks (or on an empty peg). The version shown involves three disks, but more disks can be used (making the task significantly more difficult). **Bottom:** Modified text-based version of ToH. Three lists are presented, labelled A, B and C. A set of integers is distributed amongst these lists. The goal is to move the numbers so that they are arranged in ascending order in list C. Only the number at the end of the list may be moved, and a number can only be placed in front of a smaller number. Multiple problem instances were created by varying the initial state.

## A.4 RESULTS TABLES

Table 1: Results on Valuepath task.

| Model | % solved problems | % invalid actions | Avg plan steps | | |
|---|---|---|---|---|---|
| | | | 1-step | 2-step | 4-step |
| GPT-4 Zero-shot | 54 | 8 | 2.5 | 2.5 | 5 |
| GPT-4 ICL | 91 | **0** | 1.75 | 2.33 | **4.67** |
| LLM-PFC | **100** | **0** | **1.5** | **2** | 4.75 |

Table 2: Results on Steppath task.

| Model | % solved problems | | | % invalid actions | | | Avg plan steps | | |
|---|---|---|---|---|---|---|---|---|---|
| | 2-step | 3 step | 4-step | 2-step | 3-step | 4-step | 2-step | 3-step | 4-step |
| GPT-4 Zero-shot | 75 | 40 | 20 | 9 | 13 | 18 | **2.07** | 4 | 5.25 |
| GPT-4 ICL | 74 | 74 | 42 | 10 | 6 | 14 | 2.14 | 3.78 | **4.38** |
| LLM-PFC | **100** | **100** | **95** | **0** | **0** | **0** | 2.1 | **3.42** | 4.5 |

Table 3: Results on Detour task.

| Model | % solved problems | % invalid actions |
|---|---|---|
| GPT-4 Zero-shot | 54 | 8 |
| GPT-4 ICL | 91 | **0** |
| LLM-PFC | **100** | **0** |

Table 4: Results on Reward Revaluation task.

| Model | % solved problems | % invalid actions |
|---|---|---|
| GPT-4 Zero-shot | 38 | 14 |
| GPT-4 ICL | 31 | **8** |
| LLM-PFC | **54** | **0** |

Table 5: Results on ToH. Note that we also include results here for the GPT-4 ICL baseline when prompted with 5 ICL examples (as opposed to 2 examples in the standard version of the baseline). Surprisingly, more ICL examples hurts performance on this task, perhaps due to overfitting to the specific examples (Hasanbeig et al., 2023).

| Model | % solved problems | | % invalid actions | |
|---|---|---|---|---|
| | 3-disk | 4-disk (OOD) | 3-disk | 4-disk (OOD) |
| GPT-4 Zero-shot | 11 | 2 | 30 | 50 |
| GPT-4 ICL | 46 | 1 | 12 | 41 |
| GPT-4 ICL (5 examples) | 38 | 1 | 19 | 41 |
| GPT-4 CoT + ICL | 42 | 5 | 22 | 39 |
| GPT-4 ToT | 25 | – | 4 | – |
| GPT-4 MAD | 25 | – | 24 | – |
| LLM-PFC | **74** | **24** | **0** | **0** |

Table 6: Ablation study on ToH with 3 disks.

| Model | % solved problems | % invalid actions |
|---|---|---|
| LLM-PFC | 74 | 0 |
| w/o Task Decomposer | 50 | 0 |
| w/o Tree Search | 32 | 0 |
| w/o Monitor | 27 | 31 |

## A.5 ANALYSIS OF FAILURE MODES

Failures in LLM-PFC largely stem from failures in the TaskDecomposer, Actor, and Evaluator modules. The TaskDecomposer sometimes fails to identify effective subgoals that will ultimately move the agent toward the final goal; the Actor sometimes fails to propose effective moves that move the agent toward the goal or subgoal; and the Evaluator sometimes fails to assign values that result in the selection of the best possible move as proposed by the Actor. This is in contrast to the Monitor (responsible for detection of invalid moves), Orchestrator (responsible for determining whether a goal or subgoal has been achieved), and Predictor (responsible for predicting the states that will result from a proposed action), which all perform nearly perfectly in all tasks.

## A.6 PROMPTS AND IN-CONTEXT EXAMPLES

### A.6.1 TASK DECOMPOSER

For ToH, the TaskDecomposer generated a single subgoal per problem. The in-context examples included chain-of-thought reasoning (Wei et al., 2022b) based on the goal recursion strategy (Simon, 1975) (Section A.6.1), which is sometimes provided to human participants in psychological studies of problem solving (Carpenter et al., 1990). The specific prompt and in-context examples are shown below:

```
Consider the following puzzle problem:

Problem description:
- There are three lists labeled A, B, and C.
- There is a set of numbers distributed among those three lists.
- You can only move numbers from the rightmost end of one list to
the rightmost end of another list.
Rule #1:  You can only move a number if it is at the rightmost end
of its current list.
Rule #2:  You can only move a number to the rightmost end of a
list if it is larger than the other numbers in that list.
A move is valid if it satisfies both Rule #1 and Rule #2.
A move is invalid if it violates either Rule #1 or Rule #2.

Goal:  The goal is to generate a single subgoal from the current
configuration, that helps in reaching the goal configuration using
minimum number of moves.

To generate subgoal use the goal recursion strategy.  First if the
smallest number isn't at the correct position in list C, then set
the subgoal of moving the smallest number to its correct position
in list C.But before that, the numbers larger than the smallest
number and present in the same list as the smallest number must
be moved to a list other than list C. This subgoal is recursive
because in order to move the next smallest number to the list
other than list C, the numbers larger than the next smallest
number and present in the same list as the next smallest number
must be moved to a list different from the previous other list and
so on.
```

Note in the subgoal configuration all numbers should always be in ascending order in all the three lists.

Here are two examples:

Example 1:

This is the current configuration:
A = [0,1]
B = [2]
C = []

This is the goal configuration:
A = []
B = []
C = [0, 1, 2]

Answer:
I need to move 0 from list A to list C.
Step 1.  Find the numbers to the right of 0 in list A. There is 1 to the right of 0.
Step 2.  Find the numbers larger than 0 in list C. There are none.
I will move the numbers found in Step 1 and Step 2 to list B.
Hence I will move 1 from list A to list B. Also numbers should be in ascending order in list B.
Subgoal:
A = [0]
B = [1, 2]
C = []

Example 2:

This is the current configuration:
A = [1]
B = [0]
C = [2]

This is the goal configuration:
A = []
B = []
C = [0, 1, 2]

Answer:
I need to move 0 from list B to list C.
Step 1.  Find the numbers to the right of 0 in list B. There are none.
Step 2.  Find the numbers larger than 0 in list C. There is 2 which is larger than 0.
I will move the numbers found in Step 1 and Step 2 to list A.
Hence, I will move 2 from list C to list A. Also numbers should be in ascending order in list A.
Subgoal:
A = [1, 2]
B = [0]
C = []

Here is the task:

This is the current configuration:
A = [0, 1, 2]

```
B = []
C = []

This is the goal configuration:
A = []
B = []
C = [0, 1, 2]

Answer:
```

### A.6.2 ACTOR

The $\text{Actor}$ was prompted to propose $B = 2$ distinct actions. In some instances, the $\text{Actor}$ failed to propose two distinct actions. In those cases, we iteratively scaled the temperature by a factor of 0.1. This was done for a maximum of 10 attempts or until two distinct actions were produced. If the $\text{Actor}$ was not able to propose two distinct actions even after 10 attempts, we then used only a single action. The specific prompt and in-context examples for the ToH task are shown below:

```
Consider the following puzzle problem:

Problem description:
- There are three lists labeled A, B, and C.
- There is a set of numbers distributed among those three lists.
- You can only move numbers from the rightmost end of one list to
the rightmost end of another list.
Rule #1:  You can only move a number if it is at the rightmost end
of its current list.
Rule #2:  You can only move a number to the rightmost end of a
list if it is larger than the other numbers in that list.
A move is valid if it satisfies both Rule #1 and Rule #2.
A move is invalid if it violates either Rule #1 or Rule #2.

Goal:  The goal is to end up in the goal configuration using
minimum number of moves.

Here are two examples:

Example 1:

This is the starting configuration:
A = [0, 1]
B = [2]
C = []
This is the goal configuration:
A = []
B = []
C = [0, 1, 2]

Here is the sequence of minimum number of moves to reach the goal
configuration from the starting configuration:

Move 2 from B to C.
A = [0, 1]
B = []
C = [2]

Move 1 from A to B.
A = [0]
B = [1]
C = [2]

Move 2 from C to B.
A = [0]
```

```
B = [1, 2]
C = []

Move 0 from A to C.
A = []
B = [1, 2]
C = [0]

Move 2 from B to A.
A = [2]
B = [1]
C = [0]

Move 1 from B to C.
A = [2]
B = []
C = [0, 1]

Move 2 from A to C.
A = []
B = []
C = [0, 1, 2]

Example 2:
This is the starting configuration:
A = [1]
B = [0]
C = [2]
This is the goal configuration:
A = []
B = []
C = [0, 1, 2]

Here is the sequence of minimum number of moves to reach the goal
configuration from the starting configuration:

Move 2 from C to A.
A = [1, 2]
B = [0]
C = []

Move 0 from B to C.
A = [1, 2]
B = []
C = [0]

Move 2 from A to B.
A = [1]
B = [2]
C = [0]

Move 1 from A to C.
A = []
B = [2]
C = [0, 1]

Move 2 from B to C.
A = []
B = []
C = [0, 1, 2]

Here is the task:
```

```
This is the starting configuration:
A = [0, 1, 2]
B = []
C = []
This is the goal configuration:
A = [0]
B = [1, 2]
C = []

Give me only two different valid next moves possible from the
starting configuration that would help in reaching the goal
configuration using as few moves as possible.
Your answer should be in the format as below:
1.  Move <N> from <src> to <trg>.
```

### A.6.3 MONITOR

The Monitor was prompted with chain-of-thought reasoning in which each of the rules of the task were checked before determining action validity. We stored the actions deemed valid by the Monitor in a separate buffer, and we terminated the action proposal loop (Algorithm 1) when there were two distinct actions in this buffer, or exceeded a maximum of 10 interactions with the Monitor. After termination of the action proposal loop, if the buffer didn't contain two distinct actions, we used the only action in the buffer. If the buffer was empty, we used the action(s) proposed by the Actor at the last attempt. The following text was used as a prompt and in-context examples in the ToH task:

```
Consider the following puzzle problem:

Problem description:
- There are three lists labeled A, B, and C.
- There is a set of numbers distributed among those three lists.
- You can only move numbers from the rightmost end of one list to
the rightmost end of another list.
Rule #1:  You can only move a number if it is at the rightmost end
of its current list.
Rule #2:  You can only move a number to the rightmost end of a
list if it is larger than the other numbers in that list.
A move is valid if it satisfies both Rule #1 and Rule #2.
A move is invalid if it violates either Rule #1 or Rule #2.

Goal:  The goal is to check if the proposed move satisfies or
violates Rule #1 and Rule #2 and based on that if it is a valid
or invalid move.

Here are two examples:

Example 1:

This is the initial configuration:
A = []
B = [1]
C = [0, 2]

Proposed move:
Move 0 from C to B.

Answer:
First check whether the move satisfies or violates Rule #1.  Index
of 0 in list C is 0.  Length of list C is 2.  The difference in
length of list C and index of 0 in list C is 2, which is not equal
to 1.  Hence 0 is not at the rightmost end of list C, and the move
violates Rule #1.
Next check whether the move satisfies or violates Rule #2.  For
```

that compute the maximum of list B, to which 0 is moved.  Maximum
of list B is 1.  0 is not larger than 1.  Hence the move violates
Rule #2.
Since the Move 0 from list C to list B violates both Rule #1 and
Rule #2, it is invalid.

Example 2:

This is the initial configuration:
A = []
B = [1]
C = [0, 2]

Proposed move:
Move 2 from C to B.

Answer:
First check whether the move satisfies or violates Rule #1.  Index
of 2 in list C is 1.  Length of list C is 2.  The difference in
length of list C and index of 2 in list C is 1.  Hence 2 is at the
rightmost end of list C, and the move satisfies Rule #1.
Next check whether the move satisfies or violates Rule #2.  For
that compute the maximum of list B, to which 2 is moved.  Maximum
of list B is 1.  2 is larger than 1.  Hence the move satisfies
Rule #2.
Since the Move 2 from list C to list B satisfies both Rule #1 and
Rule #2, it is valid.

Here is the task:

This is the initial configuration:
A = []
B = [0, 1]
C = [2]

Proposed move:
Move 1 from B to A.

Answer:

### A.6.4  PREDICTOR

The Predictor was prompted to predict the next state, given the current state and the proposed
action. The following text was used as a prompt and in-context examples in the ToH task:

Consider the following puzzle problem:

Problem description:
- There are three lists labeled A, B, and C.
- There is a set of numbers distributed among those three lists.
- You can only move numbers from the rightmost end of one list to
the rightmost end of another list.
Rule #1:  You can only move a number if it is at the rightmost end
of its current list.
Rule #2:  You can only move a number to the rightmost end of a
list if it is larger than the other numbers in that list.

Goal:  The goal is to predict the configuration of the
three lists, if the proposed move is applied to the current
configuration.

```
Here are two examples:

Example 1:

This is the current configuration:
A = []
B = [1]
C = [0, 2]

Proposed move:
Move 2 from list C to list B.

Answer:
A = []
B = [1, 2]
C = [0]

Example 2:

This is the current configuration:
A = []
B = [1]
C = [0, 2]

Proposed move:
Move 1 from list B to list A.

Answer:
A = [1]
B = []
C = [0, 2]

Here is the task:

This is the current configuration:
A = []
B = [0, 1]
C = [2]

Proposed move:
Move 1 from list B to list A.

Answer:
```

### A.6.5  EVALUATOR

For the Evaluator, in the ToH task, GPT-4 was prompted to generate a heuristic function that could be used to estimate the distance between the current state and the goal state. The description of this heuristic function was then included in the Evaluator's prompt. In the graph traversal tasks, we used only in-context examples, with no heuristic function prompt.

The following text shows prompt and in-context examples used in the ToH task. The first User message prompts GPT-4 to generate a heuristic function. After GPT-4's response, another User message prompts the Evaluator to use this function:

USER: `Consider the following puzzle problem:`

```
Problem description:
- There are three lists labeled A, B, and C.
- There is a set of numbers distributed among those three lists.
- You can only move numbers from the rightmost end of one list to
the rightmost end of another list.
Rule #1:  You can only move a number if it is at the rightmost end
of its current list.
```

Rule #2: You can only move a number to the rightmost end of a
list if it is larger than the other numbers in that list.
A move is valid if it satisfies both Rule #1 and Rule #2.
A move is invalid if it violates either Rule #1 or Rule #2.

Goal: The goal is to predict the minimum number of valid
moves required to reach the goal configuration from the current
configuration.

Here are two examples:

Example 1:

This is the current configuration:
A = [0, 1, 2]
B = []
C = []

This is the goal configuration:
A = []
B = []
C = [0, 1, 2]

Answer:
The minimum number of valid moves required to reach the goal
configuration from the current configuration is 7.

Example 2:

This is the current configuration:
A = [1, 2]
B = [0]
C = []

This is the goal configuration:
A = []
B = []
C = [0, 1, 2]

Answer:
The minimum number of valid moves required to reach the goal
configuration from the current configuration is 4.

What heuristic function can be used to estimate the minimum number
of valid moves required to reach the goal configuration from a
given current configuration?

GPT-4: A suitable heuristic function for this problem is the "sum
of the distances" heuristic. This heuristic estimates the minimum
number of valid moves required to reach the goal configuration by
calculating the sum of the distances each number needs to travel
to reach its final position in the goal configuration.

Here's how the heuristic function works:

1. For each number in the current configuration, determine its
current position (list and index) and its goal position (list and
index) in the goal configuration.
2. Calculate the distance between the current position and the
goal position for each number. The distance can be calculated

```
as the absolute difference between the indices of the current and
goal positions, plus a penalty if the number needs to move to a
different list.
3.  Sum the distances calculated in step 2 for all numbers.

The heuristic function will return the sum of the distances, which
is an estimate of the minimum number of valid moves required to
reach the goal configuration from the current configuration.

This heuristic is admissible because it never overestimates
the cost of reaching the goal configuration.  It considers the
minimum number of moves required for each number to reach its goal
position, without taking into account the constraints imposed by
the rules of the puzzle.  Therefore, the actual number of moves
required to reach the goal configuration will always be greater
than or equal to the heuristic value.
```

USER: `This is the current configuration:`
```
A = [0]
B = []
C = [1, 2]
This is the goal configuration:
A = [0]
B = [1, 2]
C = []

Use the heuristic function to predict the minimum number of valid
moves required to reach the goal configuration from the current
configuration.

Please provide your answer according to the heuristic function in
the format as below:
The minimum number of valid moves required to reach the goal
configuration from the current configuration is <N>.
```

### A.6.6 ORCHESTRATOR

The Orchestrator was prompted to determine whether the current subgoal was achieved by the current state. The following text was used as a prompt and in-context examples in the ToH task:

```
Consider the following puzzle problem:

Problem description:
- There are three lists labeled A, B, and C.
- There is a set of numbers distributed among those three lists.
- You can only move numbers from the rightmost end of one list to
the rightmost end of another list.
Rule #1:  You can only move a number if it is at the rightmost end
of its current list.
Rule #2:  You can only move a number to the rightmost end of a
list if it is larger than the other numbers in that list.

Goal:  The goal is to predict whether the current configuration
matches the goal configuration or not.

Here are two examples:

Example 1:

This is the current configuration:
A = []
```

```
B = []
C = [0, 1, 2]

This is the goal configuration:
A = []
B = []
C = [0, 1, 2]

Answer:  The current configuration matches the goal configuration.
Hence yes.

Example 2:

This is the current configuration:
A = [0, 1]
B = [2]
C = []

This is the goal configuration:
A = []
B = []
C = [0, 1, 2]

Answer:  The current configuration doesn't match the goal
configuration.  Hence no.

Here is the task:

This is the current configuration:
A = []
B = [0, 1, 2]
C = []

This is the goal configuration:
A = []
B = []
C = [0, 1, 2]

Answer:
```

### A.6.7 ZERO-SHOT PROMPT

An example prompt for the GPT-4 zero-shot baseline is shown below:

```
Consider the following puzzle problem:

Problem description:
- There are three lists labeled A, B, and C.
- There is a set of numbers distributed among those three lists.
- You can only move numbers from the rightmost end of one list to
the rightmost end of another list.
Rule #1:  You can only move a number if it is at the rightmost end
of its current list.
Rule #2:  You can only move a number to the rightmost end of a
list if it is larger than the other numbers in that list.
A move is valid if it satisfies both Rule #1 and Rule #2.
A move is invalid if it violates either Rule #1 or Rule #2.

Goal:  The goal is to end up in the configuration where all
numbers are in list C, in ascending order using minimum number
of moves.

This is the starting configuration:
A = [0, 1, 2]
```

```
B = []
C = []
This is the goal configuration:
A = []
B = []
C = [0,1,2]
```

Give me the sequence of moves to solve the puzzle from the starting configuration, updating the lists after each move. Please try to use as few moves as possible, and make sure to follow the rules listed above. Please limit your answer to a maximum of 10 steps.

```
Please format your answer as below:
Step 1.  Move <N> from <src> to <tgt>.
A = []
B = []
C = []
```

### A.6.8  ICL PROMPT

An example prompt for the GPT-4 ICL baseline is shown below:

```
Consider the following puzzle problem:

Problem description:
- There are three lists labeled A, B, and C.
- There is a set of numbers distributed among those three lists.
- You can only move numbers from the rightmost end of one list to
the rightmost end of another list.
Rule #1:  You can only move a number if it is at the rightmost end
of its current list.
Rule #2:  You can only move a number to the rightmost end of a
list if it is larger than the other numbers in that list.
A move is valid if it satisfies both Rule #1 and Rule #2.
A move is invalid if it violates either Rule #1 or Rule #2.

Goal:  The goal is to end up in the configuration where all
numbers are in list C, in ascending order using minimum number
of moves.

Here are two examples:

Example 1:

This is the starting configuration:
A = [0, 1]
B = [2]
C = []
This is the goal configuration:
A = []
B = []
C = [0, 1, 2]

Here is the sequence of minimum number of moves to reach the goal
configuration from the starting configuration:

Move 2 from B to C.
A = [0, 1]
B = []
C = [2]

Move 1 from A to B.
A = [0]
```

```
B = [1]
C = [2]

Move 2 from C to B.
A = [0]
B = [1, 2]
C = []

Move 0 from A to C.
A = []
B = [1, 2]
C = [0]

Move 2 from B to A.
A = [2]
B = [1]
C = [0]

Move 1 from B to C.
A = [2]
B = []
C = [0, 1]

Move 2 from A to C.
A = []
B = []
C = [0, 1, 2]

Example 2:
This is the starting configuration:
A = [1]
B = [0]
C = [2]
This is the goal configuration:
A = []
B = []
C = [0, 1, 2]

Here is the sequence of minimum number of moves to reach the goal
configuration from the starting configuration:

Move 2 from C to A.
A = [1, 2]
B = [0]
C = []

Move 0 from B to C.
A = [1, 2]
B = []
C = [0]

Move 2 from A to B.
A = [1]
B = [2]
C = [0]

Move 1 from A to C.
A = []
B = [2]
C = [0, 1]

Move 2 from B to C.
A = []
```

```
B = []
C = [0, 1, 2]

Here is the task:

This is the starting configuration:
A = [0, 1, 2]
B = []
C = []
This is the goal configuration:
A = []
B = []
C = [0,1,2]

Give me the sequence of moves to solve the puzzle from the
starting configuration, updating the lists after each move.
Please try to use as few moves as possible, and make sure to
follow the rules listed above.  Please limit your answer to a
maximum of 10 steps.

Please format your answer as below:
Step 1.  Move <N> from <src> to <tgt>.
A = []
B = []
C = []
```

### A.6.9    CoT ICL PROMPT

An example prompt for the GPT-4 CoT ICL baseline is shown below:

```
Consider the following puzzle problem:

Problem description:
- There are three lists labeled A, B, and C.
- There is a set of numbers distributed among those three lists.
- You can only move numbers from the rightmost end of one list to
the rightmost end of another list.
Rule #1:  You can only move a number if it is at the rightmost end
of its current list.
Rule #2:  You can only move a number to the rightmost end of a
list if it is larger than the other numbers in that list.
A move is valid if it satisfies both Rule #1 and Rule #2.
A move is invalid if it violates either Rule #1 or Rule #2.

Goal:  The goal is to end up in the configuration where all
numbers are in list C, in ascending order using minimum number
of moves.

Here are two examples:

Example 1:

This is the starting configuration:
A = [0, 1]
B = [2]
C = []
This is the goal configuration:
A = []
B = []
C = [0, 1, 2]
```

Here is the sequence of minimum number of moves along with
reasoning for each move to reach the goal configuration from the
starting configuration:

I need to move 0 from A to C. But before that I need to move the
number present to the right of 0, which is 1 to B. There is a
number larger than 1 already present in list B. Hence I first need
to move 2 from B to C.
Move 2 from B to C.
A = [0, 1]
B = []
C = [2]

I need to move 0 from A to C. But before that I need to move the
number present to the right of 0, which is 1 to B.
Move 1 from A to B.
A = [0]
B = [1]
C = [2]

I need to move 0 from A to C. There is a number larger than 0
already present in list C. Hence I first need to move 2 from C
to B.
Move 2 from C to B.
A = [0]
B = [1, 2]
C = []

There is no number to the right of 0 in A, and there is no number
larger than 0 in C. Hence, I can move 0 from A to C.
Move 0 from A to C.
A = []
B = [1, 2]
C = [0]

I need to move 1 from B to C. But before that I need to move the
number present to the right of 1, which is 2 to A.
Move 2 from B to A.
A = [2]
B = [1]
C = [0]

There is no number to the right of 1 in B, and there is no number
larger than 1 in C. Hence, I can move 1 from B to C.
Move 1 from B to C.
A = [2]
B = []
C = [0, 1]

There is no number to the right of 2 in A, and there is no number
larger than 2 in C. Hence, I can move 2 from A to C.
Move 2 from A to C.
A = []
B = []
C = [0, 1, 2]

Example 2:

This is the starting configuration:
A = [1]
B = [0]
C = [2]
This is the goal configuration:

```
A = []
B = []
C = [0, 1, 2]
```

Here is the sequence of minimum number of moves along with reasoning for each move to reach the goal configuration from the starting configuration:

I need to move 0 from B to C. There is a number larger than 0 already present in list C. Hence I first need to move 2 from C to A.
Move 2 from C to A.
```
A = [1, 2]
B = [0]
C = []
```

There is no number to the right of 0 in B, and there is no number larger than 0 in C. Hence, I can move 0 from B to C.
Move 0 from B to C.
```
A = [1, 2]
B = []
C = [0]
```

I need to move 1 from A to C. But before that I need to move the number present to the right of 1, which is 2 to B.
Move 2 from A to B.
```
A = [1]
B = [2]
C = [0]
```

There is no number to the right of 1 in A, and there is no number larger than 1 in C. Hence, I can move 1 from A to C.
Move 1 from A to C.
```
A = []
B = [2]
C = [0, 1]
```

There is no number to the right of 2 in B, and there is no number larger than 2 in C. Hence, I can move 2 from B to C.
Move 2 from B to C.
```
A = []
B = []
C = [0, 1, 2]
```

Here is the task:

This is the starting configuration:
```
A = [0, 1, 2]
B = []
C = []
```
This is the goal configuration:
```
A = []
B = []
C = [0,1,2]
```

Give me the sequence of moves to solve the puzzle from the starting configuration, updating the lists after each move. Please try to use as few moves as possible, and make sure to follow the rules listed above. Please limit your answer to a maximum of 10 steps.

Please format your answer as below:
Step 1. Move <N> from <src> to <tgt>.

```
A = []
B = []
C = []
```

### A.6.10   ToT GENERATOR PROMPT

To implement the generator in tree-of-thought (ToT) (Yao et al., 2023), we combined the prompts from our Actor and Predictor modules. This is because ToT operates over 'thoughts', whereas planning tasks are typically factorized into distinct actions and states. Thus, the generator was prompted to both propose actions and predict the states that would result from those actions. An example prompt is shown below:

```
Consider the following puzzle problem:

Problem description:
- There are three lists labeled A, B, and C.
- There is a set of numbers distributed among those three lists.
- You can only move numbers from the rightmost end of one list to
the rightmost end of another list.
Rule #1:  You can only move a number if it is at the rightmost end
of its current list.
Rule #2:  You can only move a number to the rightmost end of a
list if it is larger than the other numbers in that list.
A move is valid if it satisfies both Rule #1 and Rule #2.
A move is invalid if it violates either Rule #1 or Rule #2.

Goal:  The goal is to end up in the configuration where all
numbers are in list C, in ascending order using minimum number
of moves.

Here are two examples:

Example 1:

This is the starting configuration:
A = [0, 1]
B = [2]
C = []
This is the goal configuration:
A = []
B = []
C = [0, 1, 2]

Here is the sequence of minimum number of moves to reach the goal
configuration from the starting configuration:

Move 2 from B to C.
A = [0, 1]
B = []
C = [2]

Move 1 from A to B.
A = [0]
B = [1]
C = [2]

Move 2 from C to B.
A = [0]
B = [1, 2]
C = []

Move 0 from A to C.
A = []
```

```
B = [1, 2]
C = [0]

Move 2 from B to A.
A = [2]
B = [1]
C = [0]

Move 1 from B to C.
A = [2]
B = []
C = [0, 1]

Move 2 from A to C.
A = []
B = []
C = [0, 1, 2]

Example 2:
This is the starting configuration:
A = [1]
B = [0]
C = [2]
This is the goal configuration:
A = []
B = []
C = [0, 1, 2]

Here is the sequence of minimum number of moves to reach the goal
configuration from the starting configuration:

Move 2 from C to A.
A = [1, 2]
B = [0]
C = []

Move 0 from B to C.
A = [1, 2]
B = []
C = [0]

Move 2 from A to B.
A = [1]
B = [2]
C = [0]

Move 1 from A to C.
A = []
B = [2]
C = [0, 1]

Move 2 from B to C.
A = []
B = []
C = [0, 1, 2]

Starting configuration:  A = [0, 1], B = [2], C = []
Goal configuration:  A = [], B = [], C = [0, 1, 2]
Two possible valid next moves from the starting configuration that
would help in reaching the goal configuration using as few moves
as possible:
```

```
Possible move number 1:
Starting configuration:
A = [0, 1]
B = [2]
C = []
Move 2 from B to C
Current configuration:
(A = [0, 1]
B = []
C = [2])

Possible move number 2:
Starting configuration:
A = [0, 1]
B = [2]
C = []
Move 1 from A to C
Current configuration:
(A = [0]
B = [2]
C = [1])

Starting configuration:  A = [1], B = [0], C = [2]
Goal configuration:  A = [], B = [], C = [0, 1, 2]
Two possible valid next moves from the starting configuration that
would help in reaching the goal configuration using as few moves
as possible:

Possible move number 1:
Starting configuration:
A = [1]
B = [0]
C = [2]
Move 2 from C to A
Current configuration:
(A = [1, 2]
B = [0]
C = [])

Possible move number 2:
Starting configuration:
A = [1]
B = [0]
C = [2]
Move 2 from C to B
Current configuration:
(A = [1]
B = [0, 2]
C = [])

Starting configuration:  A = [0, 1, 2], B = [], C = []
Goal configuration:  A = [], B = [], C = [0, 1, 2]
Two possible valid next moves from the starting configuration that
would help in reaching the goal configuration using as few moves
as possible:

Please provide the current confguration between "(" and ")"
```

### A.6.11 ToT EVALUATOR PROMPT

To implement the evaluator in tree-of-thought (ToT) (Yao et al., 2023), we combined the prompts from our Monitor and Evaluator modules. This is because the evaluator in ToT is responsible for

performing a combination of state evaluation and error monitoring. An example prompt is shown
below:

```
Problem description:  - There are three lists labeled A, B, and C.
- There is a set of numbers distributed among those three lists.
- You can only move numbers from the rightmost end of one list to
the rightmost end of another list. Rule #1:  You can only move
a number if it is at the rightmost end of its current list. Rule
#2:  You can only move a number to the rightmost end of a list if
it is larger than the other numbers in that list.  A move is valid
if it satisfies both Rule #1 and Rule #2.  A move is invalid if it
violates either Rule #1 or Rule #2.

Goal:  The goal is to check if the proposed move satisfies or
violates Rule #1 and Rule #2 and based on that if it is a valid
or invalid move.  If the move is valid, predict the minimum number
of valid moves required to reach the goal configuration from the
current configuration.

"The following heuristic function (proposed by GPT-4) may be
helpful when estimating the the minimum number of valid moves.

GPT-4:  "A suitable heuristic function is the "sum of the
distances" heuristic.  This heuristic estimates the minimum
number of valid moves required to reach the goal configuration by
calculating the sum of the distances each number needs to travel
to reach its final position in the goal configuration.

Here's how the heuristic function works:

1.  For each number in the current configuration, determine its
current position (list and index) and its goal position (list and
index) in the goal configuration.
2.  Calculate the distance between the current position and the
goal position for each number.  The distance can be calculated
as the absolute difference between the indices of the current and
goal positions, plus a penalty if the number needs to move to a
different list.
3.  Sum the distances calculated in step 2 for all numbers.

The heuristic function will return the sum of the distances, which
is an estimate of the minimum number of valid moves required to
reach the goal configuration from the current configuration.  This
heuristic is admissible because it never overestimates the cost of
reaching the goal configuration.  It considers the minimum number
of moves required for each number to reach its goal position,
without taking into account the constraints imposed by the rules
of the puzzle.  Therefore, the actual number of moves required to
reach the goal configuration will always be greater than or equal
to the heuristic value."

Use this heuristic function to predict the minimum number of valid
moves required to reach the goal configuration from the current
configuration.

Here are two examples:

Example 1:

This is the initial configuration:
A = []
B = [1]
C = [0, 2]
```

```
Proposed move:
Move 0 from C to B.

Current configuration:
A = []
B = [1, 0]
C = [2]

Goal configuration:
A = []
B = []
C = [0, 1, 2]

Answer:
First check whether the move satisfies or violates Rule #1.  Index
of 0 in list C is 0.  Length of list C is 2.  The difference in
length of list C and index of 0 in list C is 2, which is not equal
to 1.  Hence 0 is not at the rightmost end of list C, and the move
violates Rule #1.
Next check whether the move satisfies or violates Rule #2.  For
that compute the maximum of list B, to which 0 is moved.  Maximum
of list B is 1.  0 is not larger than 1.  Hence the move violates
Rule #2.
Since the Move 0 from list C to list B violates both Rule #1 and
Rule #2, it is invalid.
Since it is an invalid move, it is impossible to reach the goal
configuration from the current configuration.

Example 2:

This is the initial configuration:
A = []
B = [1]
C = [0, 2]

Proposed move:
Move 2 from C to B.

Current configuration:
A = []
B = [1, 2]
C = [0]

Goal configuration:
A = []
B = []
C = [0, 1, 2]

Answer:
First check whether the move satisfies or violates Rule #1.  Index
of 2 in list C is 1.  Length of list C is 2.  The difference in
length of list C and index of 2 in list C is 1.  Hence 2 is at the
rightmost end of list C, and the move satisfies Rule #1.
Next check whether the move satisfies or violates Rule #2.  For
that compute the maximum of list B, to which 2 is moved.  Maximum
of list B is 1.  2 is larger than 1.  Hence the move satisfies
Rule #2.
Since the Move 2 from list C to list B satisfies both Rule #1 and
Rule #2, it is valid.
Since it is a valid move, the minimum number of valid moves
required to reach the goal configuration from the current
configuration is 3.
```

Here is the task:

This is the initial configuration:
A = [0, 1, 2]
B = []
C = []

Proposed move:
Move 2 from A to C.

Current configuration:
A = [0, 1]
B = []
C = [2]

Goal configuration:
A = []
B = []
C = [0, 1, 2]

If the proposed move is valid, use the heuristic function to predict the minimum number of valid moves required to reach the goal configuration from the current configuration.

Please provide your answer according to the heuristic function in the format as below, if it is a valid move:
The minimum number of valid moves required to reach the goal configuration from the current configuration is <N>.

Answer:

A.7   COMPUTATIONAL COST

Table 7: Average per-problem computational cost ($\pm$ the standard error of the mean) on ToH with 3 disks.

| Model | Time(s) | Num. calls | Num. input tokens | Num. output tokens |
|---|---|---|---|---|
| GPT-4 ICL | $31.73 \pm 7.2$ | $1 \pm 0.0$ | $810.88 \pm 0.1$ | $190.38 \pm 15.4$ |
| LLM-PFC | $1{,}623.53 \pm 117.4$ | $148.6 \pm 8.2$ | $109{,}090.025 \pm 6{,}567.2$ | $14{,}543.57 \pm 844.6$ |

Table 8: Average per-problem computational cost ($\pm$ the standard error of the mean) incurred by each module of LLM-PFC on ToH with 3 disks.

| Model | Time(s) | Num. calls | Num. input tokens | Num. output tokens |
|---|---|---|---|---|
| Actor | $64.03 \pm 4.4$ | $24.68 \pm 1.5$ | $38{,}274.62 \pm 2{,}796.8$ | $758.69 \pm 54.1$ |
| Monitor | $454.45 \pm 28.5$ | $46.26 \pm 2.8$ | $32{,}135.44 \pm 1{,}933.2$ | $7{,}131.85 \pm 429.6$ |
| Predictor | $43.49 \pm 2.4$ | $29.48 \pm 1.6$ | $11{,}284.65 \pm 601.3$ | $477.48 \pm 25.7$ |
| Evaluator | $357.92 \pm 21.8$ | $22.52 \pm 1.2$ | $17{,}054.15 \pm 917.6$ | $5{,}730.11 \pm 334.3$ |
| Orchestrator | $31.53 \pm 1.7$ | $24.67 \pm 1.2$ | $9{,}553.3 \pm 466.7$ | $316.0 \pm 16.0$ |
| Decomposer | $8.39 \pm 0.3$ | $1 \pm 0.0$ | $787.88 \pm 0.0$ | $129.44 \pm 2.0$ |

