# OpenReview forum: "A Prefrontal Cortex-inspired Architecture for Planning in Large Language Models"
_ICLR.cc/2024/Conference — Submitted to ICLR 2024_

### Official Review · Reviewer_i64f · 2023-11-01

**Soundness:** 2 fair
**Presentation:** 3 good
**Contribution:** 2 fair
**Rating:** 5
**Confidence:** 3

**Summary:**

This paper presents an architecture that combines multiple Large Language Models (LLMs) by drawing inspiration from the coordination observed in different sub-regions of the prefrontal cortex. The paper demonstrates the ability of the LLM-PFC architecture to successfully solve complex planning tasks when provided with prompts that correspond to the role of each sub-region of the PFC, along with a few in-context examples. This paper shows that this combined approach outperforms zero-shot and in-context learning baselines on both the graph traversal tasks and the Tower of Hanoi (ToH) tasks.

**Strengths:**

- An interesting and innovative exploration of leveraging insights from neurobiology to enhance the performance of LLMs
- The prompt engineering efforts are non-trivial

**Weaknesses:**

- The paper lacks a quantitative comparison with other approaches for enhancing LLM performance, such as some of the approaches discussed in the related work section
- The motivation and contribution of this work are a bit unclear. If the intention is to propose a method for solving planning problems, an analysis of the efficiency and the computational cost would be helpful. If the goal is to connect to the brain, more discussions about the implications of the results on brain research are expected. For example, do we observe more active coordination of sub-regions of the PFC when the subject is solving more difficult tasks? If behaviors exhibited in the PFC-LLM diverge from biological evidence, it may also be helpful to point out the distinctions.

**Questions:**

- I am a bit confused about the results shown in the middle panel of Figures 4 and 5: The PFC-LLM architecture produced zero invalid action proposals in both tasks. Does this imply that the Monitor module is unnecessary, given that its role is to identify invalid action proposals? However, this contradicts the ablation study, which demonstrates a significant drop in PFC-LLM performance without the Monitor module. Could the authors provide a little more detailed explanation of this inconsistency?
- Is it necessary to modularize each step of the planning process? Can some of the steps be combined into a single LLM for efficiency and simplicity?
- What would be some real-world applications that could benefit from this architecture?

---

> ### Author Response · Authors · 2023-11-22
> **Response to reviewer i64f**
>
> Thank you very much for the thoughtful comments and suggestions. We have provided a point-by-point reply to these issues below:
>
> ## Stronger baselines
>
> We agree that it is important to test stronger baselines that are more directly comparable to our proposed method. We have run an additional chain-of-thought (CoT) [1] baseline on the tower-of-hanoi task. We found that CoT did not improve performance over GPT-4 ICL for three-disk problems, and performed marginally better on four-disk problems, but still significantly underperformed relative to LLM-PFC. The results are presented below, and have been included in the revised manuscript (Figure 3):
>
> Results for 3-disk problems:
>
> | | % solved | % invalid |
> | ---- | ---- | ---- |
> | GPT-4 zero-shot | 11% | 30% |
> | GPT-4 ICL | 46% | 12% |
> | GPT-4 CoT | 42% | 21% |
> | LLM-PFC | **74%** | **0%** |
>
> Results for 4-disk (out-of-distribution) problems:
>
> | | % solved | % invalid |
> | ---- | ---- | ---- |
> | GPT-4 zero-shot | 2% | 50% |
> | GPT-4 ICL | 1% | 41% |
> | GPT-4 CoT | 5% | 38% |
> | LLM-PFC | **24%** | **0%** |
>
> [1] Jason Wei, Xuezhi Wang, Dale Schuurmans, Maarten Bosma, Fei Xia, Ed Chi, Quoc V Le, Denny Zhou, et al. Chain-of-thought prompting elicits reasoning in large language models. Advances in Neural Information Processing Systems, 35:24824–24837, 2022b.
>
> ## Analysis of computational cost
>
> We have now performed an analysis of the computational costs incurred by LLM-PFC, in comparison with the GPT-4 ICL baseline, on 3-disk tower-of-hanoi problems. The results are presented below in the appendix of the revised manuscript (Tables 7 and 8).
>
> ## Additional discussion of implications for neuroscience
>
> We have added the following additional discussion to Section 6 (Conclusion and Future Directions):
>
> LLM-PFC also has important implications for neuroscientific models of PFC function. Though much work has characterized the function of individual PFC subregions, there has been less emphasis on the development of integrative models in which these functions interact to carry out coordinated behavior. The present work represents a first step in that direction. An important next step will be to directly evaluate LLM-PFC as a model of neural data, which may then lead to further refinements of the model. We look forward to investigating these possibilities in future work.
>
> ## Clarification about invalid moves and monitor ablation
>
> The results in Figures 2 and 3 (% invalid) show that LLM-PFC proposes plans with zero invalid moves. This is because the invalid moves have been rejected by the Monitor, and therefore not incorporated into the final proposed plan. Thus, ablation of the Monitor results in some invalid moves being incorporated into the plan. We have revised the description of the results to clarify this (final sentence of first paragraph of Section 4)
>
> ## Necessity of modularization
>
> To address this question, we experimented with implementing LLM-PFC in a single prompt. Specifically, we created a prompt that contained 1) descriptions of each module, 2) the three algorithms describing LLM-PFC (referring to the modules described in 1) and 3) the prompts provided to all modules in LLM-PFC. GPT-4 was provided with this prompt and instructed to implement the LLM-PFC algorithm when solving a problem. We found that the response did not remotely come close to implementing the LLM-PFC algorithm, and made many errors in each step. Substantial prompt engineering and multiple examples demonstrating implementation of the algorithm would likely be necessary for this approach to be effective (if it is possible at all). We have included the prompt and an example response in the attached supplementary material.
>
> ## Potential real-world applications
>
> Many real-world tasks involve the need to develop multi-step plans subject to complex, interacting constraints. One common example can be found in the domain of logistics, e.g., coordinating resources for the efficient transportation of goods. To test whether LLM-PFC will help in these domains, we have now carried out an experiment on a benchmark involving logistics [2]. On this task, we found that GPT-4 ICL performs very poorly, only solving 10.5% (21/200) of problems correctly, whereas LLM-PFC performed significantly better, solving **31% (62/200)** of problems correctly. This demonstrates that LLM-PFC can also be beneficial in more complex, real-world settings. We have included these results in the revised manuscript.
>
> [2] Karthik Valmeekam, Matthew Marquez, Sarath Sreedharan, and Subbarao Kambhampati. On the planning abilities of large language models–a critical investigation. arXiv preprint arXiv:2305.15771, 2023.

---

> > ### Author Response · Authors · 2023-11-22
> > **Response to reviewer i64f pt.2**
> >
> > ## Additional baseline
> >
> > We have now implemented an additional tree-of-thought (ToT) baseline [3] for 3-disk Tower-of-Hanoi problems. The following passage from the revised paper describes our implementation:
> >
> > '...we evaluated tree-of-thought (ToT) (Yao et al., 2023). Similar to LLM-PFC, ToT combines multiple LLM modules – a generator and an evaluator – to perform tree search. We implemented the generator by combining the prompts from our Actor and Predictor modules, and implemented the evaluator by combining the prompts from our Monitor and Evaluator modules (see
> > Sections A.6.10-A.6.11). We used the codebase provided by Yao et al. (2023). Plans were terminated when the predicted state matched the goal state (based on a groundtruth evaluation, as opposed to requiring the model to make this determination for itself as in LLM-PFC). For each problem, we selected the best out of five proposed plans (again based on a groundtruth evaluation). Although these decisions arguably gave ToT an advantage relative to LLM-PFC, we chose to evaluate ToT in
> > this way so as to give it the best possible chance of performing well in our task setting.'
> >
> > We found that ToT significantly underperformed on this task, solving only 25% of problems (as opposed to 74% in LLM-PFC), and proposing invalid actions 4% of the time (as opposed to 0% in LLM-PFC). This is likely due to the fact that each module is responsible for carrying out multiple functions, increasing the likelihood of errors. ToT especially tended to make errors in monitoring (error detection) and action proposal. This emphasizes the benefit of more extensively factorizing the planning procedure as in LLM-PFC. These results are now presented in Figure 3.
> >
> > [3] Yao, S., Yu, D., Zhao, J., Shafran, I., Griffiths, T. L., Cao, Y., & Narasimhan, K. (2023). Tree of thoughts: Deliberate problem solving with large language models. arXiv preprint arXiv:2305.10601.
> >
> > We would like to thank the reviewer once again for their helpful suggestions and comments. Please let us know if there are any remaining issues to address.

---

> > > ### Author Response · Authors · 2023-11-23
> > > **Response to reviewer i64f pt.3**
> > >
> > > ## Additional baseline
> > >
> > > We have added another baseline, multi-agent debate [4]. In this baseline, a solution is generated through debate between multiple LLM instances (each an instance of the GPT-4 ICL baseline), but these instances are not specialized to perform specific functions. Thus, the baseline tests for the extent to which LLM-PFC's planning performance depends specifically on the specialized nature of the modules vs. interaction between multiple instances in general. This baseline also underperformed on tower of hanoi, solving only 25% of problems correctly (vs. 74% for LLM-PFC). We have added this result to Figure 3.
> > >
> > > [4] Du, Y., Li, S., Torralba, A., Tenenbaum, J. B., & Mordatch, I. (2023). Improving Factuality and Reasoning in Language Models through Multiagent Debate. arXiv preprint arXiv:2305.14325.
> > >
> > > Thank you again for the helpful suggestions and comments. We hope that the concerns raised in the initial review have been addressed.

---

### Official Review · Reviewer_upo6 · 2023-11-01

**Soundness:** 4 excellent
**Presentation:** 4 excellent
**Contribution:** 3 good
**Rating:** 5
**Confidence:** 4

**Summary:**

The paper introduces LLM-PFC, a novel method utilizing black box large language models (LLMs) to address planning problems. Inspired by the prefrontal cortex, LLM-PFC consists of a task decomposer, actor, monitor, predictor, evaluator, and task coordinator submodules for decomposing planning problems. The method demonstrates impressive proficiency in multi-step planning, particularly in graph traversal and Tower of Hanoi tasks.

**Strengths:**

- LLM-PFC outperforms a GPT-4 baseline in several planning problems. Furthermore, the paper analyzes the importance of the components of LLM-PFC.
- LLM-PFC helps overcome hallucinations in planning problems, demonstrated by the method not outputting invalid actions for either of the considered domains.
- The paper clearly describes the LLM-PFC submodules and how they interact.
- The method is easily reproducible since the paper describes the prompts and hyperparameters for the submodules.

**Weaknesses:**

- LLM-PFC is a general reasoning and planning method. However, the work only evaluates LLM-PFC in three problems (Valuepath, Steppath, and Tower of Hanoi). Results on additional domains are needed to confirm the usefulness of the method. Even within the domain of the CogEval protocol, the paper states, "there are more challenging planning tasks (including shortcuts and detour)." Why does the paper exclude these harder problems, especially given that LLM-PFC is a zero-shot method? Even if LLM-PFC doesn't perform as well in these harder problems, these results would still provide valuable insight into where LLM-PFC fails.
- The related work section describes several related approaches that use intermediate computations with black box LLMs, such as scratchpads, chain-of-thought, tree-of-thoughts, reflexion, Society of Mind, and Describe-Explain-Plan-Select. The paper does not compare to these methods in the experiments section, even though it appears these methods are directly comparable to LLM-PFC. This makes it difficult to assess the empirical strengths of LLM-PFC over these prior works.
- The paper needs a more precise characterization of how LLM-PFC relates to prior work. Section 5 states that LLM-PFC shares some components with prior black box approaches but introduces new components and combines components in a novel way. Which components are shared by which prior works? How does LLM-PFC combine these components in a novel manner?
- While LLM-PFC achieves near-perfect results on Valuepath and Steppath, and outperforms baselines in ToH, what types of failures does it encounter in ToH? The paper should analyze the LLM-PFC failure modes and which components are responsible for failures.
- Minor: I suggest including y-axis lines in the result figures (Fig 4, 5) to easier see what values the bars correspond to (even with the full values in Appendix A.1).

The lack of experimental domains demonstrating the applicability of LLM-PFC, along with missing baselines from prior work, are the primary reasons for my final score selection.

**Questions:**

- In Fig. 4, is there any insight into why baselines and LLM-PFC achieve similar step counts for successful trajectories?

---

> ### Author Response · Authors · 2023-11-22
> **Response to reviewer upo6**
>
> Thank you very much for the thoughtful comments and suggestions. We have provided a point-by-point reply to these issues below:
>
> ## Need for more extensive evaluation
>
> We agree that a more comprehensive set of tests is important for establishing the generality of our approach. We have now evaluated the model on two additional graph traversal tasks from the CogEval benchmark [1], both of which are more difficult than the graph traversal tasks included in the initial submission. In one of these tasks, ‘Detour’, the system is first prompted to solve a single ValuePath problem (given an initial graph state and two in-context examples). After solving this problem, a ‘detour’ is then added to the graph, by removing one edge and adding a new one. The system is then prompted to solve a problem on the updated graph (with graph description, in-context examples, and the solved problem from the previous graph as context). In another task, ‘Reward Revaluation’, after solving one problem from the ValuePath task, the reward structure of the task is altered (the rewards assigned to the two potential reward locations are changed, but the graph structure remains the same) and the system is prompted to solve a new problem. Both of these tasks test the ability to flexibly generalize to new circumstances, and are more difficult than the standard ValuePath task. We find that LLM-PFC significantly outperforms GPT-4 zero-shot and GPT-4 ICL (with in-context learning examples) on both of these tasks. The results are presented below:
>
> Results for Detour task:
>
> | | % solved | % invalid |
> | ---- | ---- | ---- |
> | GPT-4 zero-shot | 54% | 6% |
> | GPT-4 ICL | 38% | 3% |
> | LLM-PFC | **92%** | **0%** |
>
> Results for Reward Revaluation task:
>
> | | % solved | % invalid |
> | ---- | ---- | ---- |
> | GPT-4 zero-shot | 38% | 14% |
> | GPT-4 ICL | 31% | 8% |
> | LLM-PFC | **54%** | **0%** |
>
> We have also included these results in the revised manuscript (Figure 2).
>
> We have also carried out an experiment on a more challenging real-world planning benchmark involving logistics (e.g., transportation of goods) [2]. On this task (we specifically focused on the ‘plan generation’ task from [2]), we found that GPT-4 ICL performs very poorly, only solving 10.5% (21/200) of problems correctly, whereas LLM-PFC performed significantly better, solving **31% (62/200)** of problems correctly. This demonstrates that LLM-PFC can also be beneficial in more complex, real-world settings. We have included these results in the revised manuscript.
>
> [1] Ida Momennejad, Hosein Hasanbeig, Felipe Vieira Frujeri, Hiteshi Sharma, Robert Osazuwa Ness, Nebojsa Jojic, Hamid Palangi, and Jonathan Larson. Evaluating cognitive maps in large language models with cogeval: No emergent planning. In Advances in neural information processing systems, volume 37, 2023. URL: https://arxiv.org/abs/2309.15129.
>
> [2] Karthik Valmeekam, Matthew Marquez, Sarath Sreedharan, and Subbarao Kambhampati. On the planning abilities of large language models–a critical investigation. arXiv preprint arXiv:2305.15771, 2023.
>
> ## Stronger baselines
>
> We also agree that it is important to test stronger baselines that are more directly comparable to our proposed method. We have run an additional chain-of-thought (CoT) [3] baseline on the tower-of-hanoi task. We found that CoT did not improve performance over GPT-4 ICL for three-disk problems, and performed marginally better on four-disk problems, but still significantly underperformed relative to LLM-PFC. The results are presented below, and have been included in the revised manuscript (Figure 3):
>
> Results for 3-disk problems:
>
> | | % solved | % invalid |
> | ---- | ---- | ---- |
> | GPT-4 zero-shot | 11% | 30% |
> | GPT-4 ICL | 46% | 12% |
> | GPT-4 CoT | 42% | 21% |
> | LLM-PFC | **74%** | **0%** |
>
> Results for 4-disk (out-of-distribution) problems:
>
> | | % solved | % invalid |
> | ---- | ---- | ---- |
> | GPT-4 zero-shot | 2% | 50% |
> | GPT-4 ICL | 1% | 41% |
> | GPT-4 CoT | 5% | 38% |
> | LLM-PFC | **24%** | **0%** |
>
> [3] Jason Wei, Xuezhi Wang, Dale Schuurmans, Maarten Bosma, Fei Xia, Ed Chi, Quoc V Le, Denny Zhou, et al. Chain-of-thought prompting elicits reasoning in large language models. Advances in Neural Information Processing Systems, 35:24824–24837, 2022b.

---

> ### Author Response · Authors · 2023-11-22
> **Response to reviewer upo6 pt.2**
>
> ## More precise characterization of relationship between LLM-PFC and prior work
>
> LLM-PFC can be contrasted with prior work in the following ways:
> - Similar to LLM-PFC, both scratchpad and chain-of-thought (CoT) decompose a problem into intermediate computations. However, unlike LLM-PFC, neither scratchpad nor CoT factorize these intermediate computations into specialized modules.
> - Tree-of-thought (ToT) introduces some degree of factorization, but the factorization is not as extensive as in LLM-PFC. The ‘generator’ module in ToT carries out a combination of the functions carried out by both the Actor (action proposal) and the Predictor (prediction of the states that will result from these actions) in LLM-PFC. The ‘evaluator’ module in ToT carries out a combination of the functions carried out by both the Monitor (error detection) and the Evaluator (prediction of state value) in LLM-PFC. ToT does not contain any component that carries out the functions of the Task Decomposer (subgoal proposal) and the Orchestrator (autonomously determining when a goal or subgoal has been achieved).
> - Multi-agent debate (i.e., Society of Mind) involves the interaction of multiple LLM instances; but, unlike LLM-PFC, these model instances are not specialized to perform specific functions.
> - Similar to LLM-PFC, reflexion involves an element of self evaluation of proposed policies, but this depends on interaction with the external environment to determine the outcome of each policy. In LLM-PFC, this self evaluation process is entirely internal to the agent.
> - Describe-Explain-Plan-Select involves the coordination of multiple modules, but the approach is specific to settings involving an agent that is spatially embedded in a 2D environment. For instance, the method utilizes the spatial proximity of objects to the agent for prioritization of subgoals.
>
> We have included this extended discussion of related work in the Appendix (Section A.1).
>
> ## Analysis of failure modes
>
> We find that failures in LLM-PFC largely stem from failures in the TaskDecomposer, Actor, and Evaluator modules. The TaskDecomposer sometimes fails to identify effective subgoals that will ultimately move the agent toward the final goal; the Actor sometimes fails to propose effective moves that move the agent toward the goal or subgoal; and the Evaluator sometimes fails to assign values that result in the selection of the best possible move as proposed by the Actor. This is in contrast to the Monitor (responsible for detection of invalid moves), Orchestrator (previously referred to as ‘TaskCoordinator’; responsible for determining whether a goal or subgoal has been achieved), and Predictor (responsible for predicting the states that will result from a proposed action), which all perform nearly perfectly in all tasks. We have now included additional discussion of these points in the appendix of the revised paper (Section A.5).
>
> ## Including y-axis lines in figures
>
> Thank you for this suggestion. We have made this change (Figures 2 and 3).
>
> ## In Fig. 4, is there any insight into why baselines and LLM-PFC achieve similar step counts for successful trajectories?
>
> Note that this is only the case for 2-step problems, which are the simplest problems on this task. For the more difficult 3-step and 4-step problems, the baselines take more steps to solve the problem (in addition to solving fewer problems).

---

> > ### Author Response · Authors · 2023-11-22
> > **Response to reviewer upo6 pt.3**
> >
> > ## Additional baseline
> >
> > We have now implemented an additional tree-of-thought (ToT) baseline [4] for 3-disk Tower-of-Hanoi problems. The following passage from the revised paper describes our implementation:
> >
> > '...we evaluated tree-of-thought (ToT) (Yao et al., 2023). Similar to LLM-PFC, ToT combines multiple LLM modules – a generator and an evaluator – to perform tree search. We implemented the generator by combining the prompts from our Actor and Predictor modules, and implemented the evaluator by combining the prompts from our Monitor and Evaluator modules (see
> > Sections A.6.10-A.6.11). We used the codebase provided by Yao et al. (2023). Plans were terminated when the predicted state matched the goal state (based on a groundtruth evaluation, as opposed to requiring the model to make this determination for itself as in LLM-PFC). For each problem, we selected the best out of five proposed plans (again based on a groundtruth evaluation). Although these decisions arguably gave ToT an advantage relative to LLM-PFC, we chose to evaluate ToT in
> > this way so as to give it the best possible chance of performing well in our task setting.'
> >
> > We found that ToT significantly underperformed on this task, solving only 25% of problems (as opposed to 74% in LLM-PFC), and proposing invalid actions 4% of the time (as opposed to 0% in LLM-PFC). This is likely due to the fact that each module is responsible for carrying out multiple functions, increasing the likelihood of errors. ToT especially tended to make errors in monitoring (error detection) and action proposal. This emphasizes the benefit of more extensively factorizing the planning procedure as in LLM-PFC. These results are now presented in Figure 3.
> >
> > [4] Yao, S., Yu, D., Zhao, J., Shafran, I., Griffiths, T. L., Cao, Y., & Narasimhan, K. (2023). Tree of thoughts: Deliberate problem solving with large language models. arXiv preprint arXiv:2305.10601.
> >
> > We would like to thank the reviewer once again for their helpful suggestions and comments. Please let us know if there are any remaining issues to address.

---

> > > ### Author Response · Authors · 2023-11-23
> > > **Response to reviewer upo6 pt.4**
> > >
> > > ## Additional baseline
> > >
> > > We have added another baseline, multi-agent debate [5]. In this baseline, a solution is generated through debate between multiple LLM instances (each an instance of the GPT-4 ICL baseline), but these instances are not specialized to perform specific functions. Thus, the baseline tests for the extent to which LLM-PFC's planning performance depends specifically on the specialized nature of the modules vs. interaction between multiple instances in general. This baseline also underperformed on tower of hanoi, solving only 25% of problems correctly (vs. 74% for LLM-PFC). We have added this result to Figure 3.
> > >
> > > [5] Du, Y., Li, S., Torralba, A., Tenenbaum, J. B., & Mordatch, I. (2023). Improving Factuality and Reasoning in Language Models through Multiagent Debate. arXiv preprint arXiv:2305.14325.
> > >
> > > Thank you again for the helpful suggestions and comments. We hope that the concerns raised in the initial review have been addressed.

---

### Official Review · Reviewer_PrdT · 2023-11-02

**Soundness:** 2 fair
**Presentation:** 3 good
**Contribution:** 2 fair
**Rating:** 5
**Confidence:** 3

**Summary:**

LLMs often struggle with tasks that require multi-step reasoning and planning. To address this, the authors propose an architecture composed of multiple interacting LLM-based modules inspired by the prefrontal cortex. Each individual module in this architecture is an instance of an LLM constructed through a combination of prompting and in-context learning and has a dedicated role (e.g., the task decomposer breaks down the high-level goal into a sequence of sub-goals). This combined architecture is evaluated on two planning tasks: graph traversal and Tower of Hanoi.

**Strengths:**

The paper introduces a new approach that combines multiple LLM instances to tackle problems requiring multi-step reasoning and planning. This architecture, which leverages insights from neuroscience, is interesting. The presentation is clear and ensures that the paper is easily comprehensible.

**Weaknesses:**

The experimental evaluation and results are not entirely convincing. In the Valuepath task, GPT-4 ICL performs nearly as well as LLM-PFC. In the Steppath task, the performance of GPT-4 ICL is comparable, except in the 4-step case.

Tower of Hanoi is a harder problem, yet ICL achieves approximately 50% success in the 3-disk case. In the 4-disk case, both zero-shot and ICL performance is nearly 0, but even the combined architecture only reaches about ~25% success. The authors do acknowledge this in their conclusion.

Minor:
- The y-axis tick labels for plots showing %solved (/invalid) should range from 0 to 100, rather than 0 to 1
- Typo on line 1 of introduction - Devlin et al., 2090

**Questions:**

- What prompts were used for GPT-4 zero-shot and ICL settings? To what extent does performance rely on the specific prompts used, and how much contextual information did these prompts provide in each case?
- If a greater number of ICL examples were utilized, would performance improvements potentially allow for a match with the combined architecture in at least a subset of the tasks?
- It appears that the problem descriptions in the prompts for each module are identical. What would be the impact on performance if these descriptions were slightly rephrased for each module?
- In the evaluation of the Tower of Hanoi task, have you experimented with prompts that incorporate different lists (e.g., X,Y,Z) and numbers than those used in constructing the individual modules?

---

> ### Author Response · Authors · 2023-11-22
> **Response to reviewer PrdT**
>
> Thank you very much for the thoughtful comments and suggestions. We have provided a point-by-point reply to these issues below:
>
> ## More difficult graph traversal tasks:
>
> The reviewer notes that the GPT-4 ICL baseline performs nearly as well as LLM-PFC on the two graph traversal tasks in the initial submission (ValuePath and StepPath). To address this, we have now evaluated the model on two additional graph traversal tasks from the CogEval benchmark [1], both of which are more difficult than the graph traversal tasks included in the initial submission. In one of these tasks, ‘Detour’, the system is first prompted to solve a single ValuePath problem (given an initial graph state and two in-context examples). After solving this problem, a ‘detour’ is then added to the graph, by removing one edge and adding a new one. The system is then prompted to solve a problem on the updated graph (with graph description, in-context examples, and the solved problem from the previous graph as context). In another task, ‘Reward Revaluation’, after solving one problem from the ValuePath task, the reward structure of the task is altered (the rewards assigned to the two potential reward locations are changed, but the graph structure remains the same) and the system is prompted to solve a new problem. Both of these tasks test the ability to flexibly generalize to new circumstances, and are more difficult than the standard ValuePath task. We find that LLM-PFC significantly outperforms GPT-4 zero-shot and GPT-4 ICL on both of these tasks. The results are presented below:
>
> Results for Detour task:
>
> | | % solved | % invalid |
> | ---- | ---- | ---- |
> | GPT-4 zero-shot | 54% | 6% |
> | GPT-4 ICL | 38% | 3% |
> | LLM-PFC | **92%** | **0%** |
>
> Results for Reward Revaluation task:
>
> | | % solved | % invalid |
> | ---- | ---- | ---- |
> | GPT-4 zero-shot | 38% | 14% |
> | GPT-4 ICL | 31% | 8% |
> | LLM-PFC | **54%** | **0%** |
>
> We have also included these results in the revised manuscript (Figure 2).
>
> [1] Ida Momennejad, Hosein Hasanbeig, Felipe Vieira Frujeri, Hiteshi Sharma, Robert Osazuwa Ness, Nebojsa Jojic, Hamid Palangi, and Jonathan Larson. Evaluating cognitive maps in large language models with cogeval: No emergent planning. In Advances in neural information processing systems, volume 37, 2023. URL: https://arxiv.org/abs/2309.15129.
>
> ## Y-axis tick labels and typo in reference
>
> Thank you for pointing out these errors. We have now implemented these changes in the revised manuscript (Figures 2 and 3).
>
> ## What prompts were used for GPT-4 zero-shot and ICL settings? To what extent does performance rely on the specific prompts used, and how much contextual information did these prompts provide in each case?
>
> First, we note that the modules each received the same number of in-context examples as the GPT-4 ICL baseline (either two or three, depending on the task). We have also now included example prompts used for the zero-shot and ICL baselines in the appendix of the revised paper (Sections A.6.7 and A.6.8). We have also included an example prompt for the chain-of-thought baseline added during the rebuttal period (Section A.6.9).
>
> Second, we have experimented with implementing LLM-PFC in a single prompt, and found that this approach was completely incapable of solving even a single problem. Specifically, we created a prompt that contained 1) descriptions of each module, 2) the three algorithms describing LLM-PFC (referring to the modules described in 1) and 3) the prompts provided to all modules in LLM-PFC. GPT-4 was provided with this prompt and instructed to implement the LLM-PFC algorithm when solving a problem. We found that the response did not remotely come close to implementing the LLM-PFC algorithm, and made many errors in each step. Substantial prompt engineering and multiple examples demonstrating implementation of the algorithm would likely be necessary for this approach to be effective (if it is possible at all). But this experiment demonstrates that the specific prompts per se are not adequate for achieving performance on par with LLM-PFC. The factorization into distinct modules and symbolic execution of the LLM-PFC algorithm is necessary to achieve strong performance. We have included the prompt and an example response in the attached supplementary material.

---

> ### Author Response · Authors · 2023-11-22
> **Response to reviewer PrdT pt.2**
>
> ## If a greater number of ICL examples were utilized, would performance improvements potentially allow for a match with the combined architecture in at least a subset of the tasks?
>
> We carried out an experiment to test this, and found that the GPT-4 ICL baseline surprisingly performs worse with more examples (5 vs. 2 examples in the original submission). We believe that this is because GPT-4 overfits to the additional ICL examples, rather than learning to carry out the planning process (a phenomenon that has also been observed in other work [2]). We present the results below, and have also included them in the appendix of the revised paper (Table 5):
>
> Results for 3-disk problems:
>
> | | % solved | % invalid |
> | ---- | ---- | ---- |
> | GPT-4 zero-shot | 11% | 30% |
> | GPT-4 ICL (2 examples) | 46% | 12% |
> | GPT-4 ICL (5 examples) | 38% | 19% |
> | LLM-PFC | **74%** | **0%** |
>
> Results for 4-disk (out-of-distribution) problems:
>
> | | % solved | % invalid |
> | ---- | ---- | ---- |
> | GPT-4 zero-shot | 2% | 50% |
> | GPT-4 ICL (2 examples) | 1% | 41% |
> | GPT-4 ICL (5 examples) | 1% | 41% |
> | LLM-PFC | **24%** | **0%** |
>
>  [2] Hasanbeig, H., Sharma, H., Betthauser, L., Frujeri, F. V., & Momennejad, I. (2023). ALLURE: A Systematic Protocol for Auditing and Improving LLM-based Evaluation of Text using Iterative In-Context-Learning. arXiv preprint arXiv:2309.13701.
>
> ## It appears that the problem descriptions in the prompts for each module are identical. What would be the impact on performance if these descriptions were slightly rephrased for each module?
>
> We chose to use identical problem descriptions for each module because this was the most straightforward implementation, and would enable LLM-PFC to be straightforwardly extended to other tasks (rather than requiring the prompts to be optimized for each module in a task-specific manner). We do not expect that the specific problem description should have an impact on LLM-PFC’s performance, so long as the problem is clearly described.

---

> > ### Author Response · Authors · 2023-11-22
> > **Response to reviewer PrdT pt.3**
> >
> > ## Additional baseline
> >
> > We have now implemented an additional tree-of-thought (ToT) baseline [3] for 3-disk Tower-of-Hanoi problems. The following passage from the revised paper describes our implementation:
> >
> > '...we evaluated tree-of-thought (ToT) (Yao et al., 2023). Similar to LLM-PFC, ToT combines multiple LLM modules – a generator and an evaluator – to perform tree search. We implemented the generator by combining the prompts from our Actor and Predictor modules, and implemented the evaluator by combining the prompts from our Monitor and Evaluator modules (see
> > Sections A.6.10-A.6.11). We used the codebase provided by Yao et al. (2023). Plans were terminated when the predicted state matched the goal state (based on a groundtruth evaluation, as opposed to requiring the model to make this determination for itself as in LLM-PFC). For each problem, we selected the best out of five proposed plans (again based on a groundtruth evaluation). Although these decisions arguably gave ToT an advantage relative to LLM-PFC, we chose to evaluate ToT in
> > this way so as to give it the best possible chance of performing well in our task setting.'
> >
> > We found that ToT significantly underperformed on this task, solving only 25% of problems (as opposed to 74% in LLM-PFC), and proposing invalid actions 4% of the time (as opposed to 0% in LLM-PFC). This is likely due to the fact that each module is responsible for carrying out multiple functions, increasing the likelihood of errors. ToT especially tended to make errors in monitoring (error detection) and action proposal. This emphasizes the benefit of more extensively factorizing the planning procedure as in LLM-PFC. These results are now presented in Figure 3.
> >
> > [3] Yao, S., Yu, D., Zhao, J., Shafran, I., Griffiths, T. L., Cao, Y., & Narasimhan, K. (2023). Tree of thoughts: Deliberate problem solving with large language models. arXiv preprint arXiv:2305.10601.
> >
> > We would like to thank the reviewer once again for their helpful suggestions and comments. Please let us know if there are any remaining issues to address.

---

> > > ### Author Response · Authors · 2023-11-23
> > > **Response to reviewer PrdT pt.4**
> > >
> > > ## Additional baseline
> > >
> > > We have added another baseline, multi-agent debate [4]. In this baseline, a solution is generated through debate between multiple LLM instances (each an instance of the GPT-4 ICL baseline), but these instances are not specialized to perform specific functions. Thus, the baseline tests for the extent to which LLM-PFC's planning performance depends specifically on the specialized nature of the modules vs. interaction between multiple instances in general. This baseline also underperformed on tower of hanoi, solving only 25% of problems correctly (vs. 74% for LLM-PFC). We have added this result to Figure 3.
> > >
> > > [4] Du, Y., Li, S., Torralba, A., Tenenbaum, J. B., & Mordatch, I. (2023). Improving Factuality and Reasoning in Language Models through Multiagent Debate. arXiv preprint arXiv:2305.14325.
> > >
> > > Thank you again for the helpful suggestions and comments. We hope that the concerns raised in the initial review have been addressed.

---

### Official Review · Reviewer_t84f · 2023-11-06

**Soundness:** 3 good
**Presentation:** 3 good
**Contribution:** 2 fair
**Rating:** 6
**Confidence:** 4

**Summary:**

This paper presents a structure of interacting LLMs. The particular structure is organised to match the believed roles of different parts of PFC. The LLMs are wrapped in an overall algorithm that prescribes the role of each LLM, how they talk to each other, and when to stop searching for an answer etc. The way each LLM knows its role is via a specific prompt, and communications from the other LLMs are appended to the prompt. The aim is that the particular structure of interacting LLMs can inherit some of the multi-step planning abilities of PFC. Two tasks are presented - a graph traversal task and a tower of hanoi task - that are hard for individual LLMs to solve. The LLM-PFC improves performance on both tasks.

**Strengths:**

The paper is very clearly presented. The model is an interesting attempt at integrating our understanding from cognitive and neuroscience into LLMs. The results on the two tasks improve upon GPT on its own.

**Weaknesses:**

Only two tasks are presented. These are a fair distance from a wide range of tasks that a general learner can solve. Given the huge computational resources to train GPT and the giant model that it is, these are really very tiny tasks to tackle with such big models. There would need to be a *much* more impressive demonstration of this technique. It’s just way too early to claim this as anything close to a general mechanism, or show that the LLM-PFC is a sensible approach

The baselines are really quite hindered compared to the proposed model. What happens when you prompt a simple GPT with the whole PFC setup? I.e. tell it that it is a PFC with all these components etc. I.e. can we tease apart the role of a prompt versus the role of the actual architecture of interacting LLMs…

**Questions:**

See weaknesses

---

> ### Author Response · Authors · 2023-11-22
> **Response to reviewer t84f**
>
> Thank you very much for the thoughtful comments and suggestions. We have provided a point-by-point reply to these issues below:
>
> ## Need for more extensive evaluation
>
> We agree that a more comprehensive set of tests is important for establishing the generality of our approach. We have now evaluated the model on two additional graph traversal tasks from the CogEval benchmark [1], both of which are more difficult than the graph traversal tasks included in the initial submission. In one of these tasks, ‘Detour’, the system is first prompted to solve a single ValuePath problem (given an initial graph state and two in-context examples). After solving this problem, a ‘detour’ is then added to the graph, by removing one edge and adding a new one. The system is then prompted to solve a problem on the updated graph (with graph description, in-context examples, and the solved problem from the previous graph as context). In another task, ‘Reward Revaluation’, after solving one problem from the ValuePath task, the reward structure of the task is altered (the rewards assigned to the two potential reward locations are changed, but the graph structure remains the same) and the system is prompted to solve a new problem. Both of these tasks test the ability to flexibly generalize to new circumstances, and are more difficult than the standard ValuePath task. We find that LLM-PFC significantly outperforms GPT-4 zero-shot and GPT-4 ICL (with in-context learning examples) on both of these tasks. The results are presented below:
>
> Results for Detour task:
>
> | | % solved | % invalid |
> | ---- | ---- | ---- |
> | GPT-4 zero-shot | 54% | 6% |
> | GPT-4 ICL | 38% | 3% |
> | LLM-PFC | **92%** | **0%** |
>
> Results for Reward Revaluation task:
>
> | | % solved | % invalid |
> | ---- | ---- | ---- |
> | GPT-4 zero-shot | 38% | 14% |
> | GPT-4 ICL | 31% | 8% |
> | LLM-PFC | **54%** | **0%** |
>
> We have also included these results in the revised manuscript (Figure 2).
>
> We have also carried out an experiment on a more challenging real-world planning benchmark involving logistics (e.g., transportation of goods) [2]. On this task (we specifically focused on the ‘plan generation’ task from [2]), we found that GPT-4 ICL performs very poorly, only solving 10.5% (21/200) of problems correctly, whereas LLM-PFC performed significantly better, solving **31% (62/200)** of problems correctly. This demonstrates that LLM-PFC can also be beneficial in more complex, real-world settings. We have included these results in the revised manuscript.
>
> [1] Ida Momennejad, Hosein Hasanbeig, Felipe Vieira Frujeri, Hiteshi Sharma, Robert Osazuwa Ness, Nebojsa Jojic, Hamid Palangi, and Jonathan Larson. Evaluating cognitive maps in large language models with cogeval: No emergent planning. In Advances in neural information processing systems, volume 37, 2023. URL: https://arxiv.org/abs/2309.15129.
>
> [2] Valmeekam, K., Marquez, M., Sreedharan, S., & Kambhampati, S. (2023). On the Planning Abilities of Large Language Models--A Critical Investigation. arXiv preprint arXiv:2305.15771.

---

> ### Author Response · Authors · 2023-11-22
> **Response to reviewer t84f pt.2**
>
> ## Stronger baselines
>
> We also agree that it is important to test stronger baselines that are more directly comparable to our proposed method. We have addressed this concern in two ways. First, we have run an additional chain-of-thought (CoT) [3] baseline on the tower-of-hanoi task. We found that CoT did not improve performance over GPT-4 ICL for three-disk problems, and performed marginally better on four-disk problems, but still significantly underperformed relative to LLM-PFC. The results are presented below, and have been included in the revised manuscript (Figure 3):
>
> Results for 3-disk problems:
>
> | | % solved | % invalid |
> | ---- | ---- | ---- |
> | GPT-4 zero-shot | 11% | 30% |
> | GPT-4 ICL | 46% | 12% |
> | GPT-4 CoT | 42% | 21% |
> | LLM-PFC | **74%** | **0%** |
>
> Results for 4-disk (out-of-distribution) problems:
>
> | | % solved | % invalid |
> | ---- | ---- | ---- |
> | GPT-4 zero-shot | 2% | 50% |
> | GPT-4 ICL | 1% | 41% |
> | GPT-4 CoT | 5% | 38% |
> | LLM-PFC | **24%** | **0%** |
>
> Second, we have experimented with implementing LLM-PFC in a single prompt. Specifically, we created a prompt that contained 1) descriptions of each module, 2) the three algorithms describing LLM-PFC (referring to the modules described in 1) and 3) the prompts provided to all modules in LLM-PFC. GPT-4 was provided with this prompt and instructed to implement the LLM-PFC algorithm when solving a problem. We found that the response did not remotely come close to implementing the LLM-PFC algorithm, and made many errors in each step. Substantial prompt engineering and multiple examples demonstrating implementation of the algorithm would likely be necessary for this approach to be effective (if it is possible at all). We have included the prompt and an example response in the attached supplementary material.
>
> [3] Jason Wei, Xuezhi Wang, Dale Schuurmans, Maarten Bosma, Fei Xia, Ed Chi, Quoc V Le, Denny Zhou, et al. Chain-of-thought prompting elicits reasoning in large language models. Advances in Neural Information Processing Systems, 35:24824–24837, 2022b.

---

> > ### Author Response · Authors · 2023-11-22
> > **Response to reviewer t84f pt.3**
> >
> > ## Additional baseline
> >
> > We have now implemented an additional tree-of-thought (ToT) baseline [4] for 3-disk Tower-of-Hanoi problems. The following passage from the revised paper describes our implementation:
> >
> > '...we evaluated tree-of-thought (ToT) (Yao et al., 2023). Similar to LLM-PFC, ToT combines multiple LLM modules – a generator and an evaluator – to perform tree search. We implemented the generator by combining the prompts from our Actor and Predictor modules, and implemented the evaluator by combining the prompts from our Monitor and Evaluator modules (see
> > Sections A.6.10-A.6.11). We used the codebase provided by Yao et al. (2023). Plans were terminated when the predicted state matched the goal state (based on a groundtruth evaluation, as opposed to requiring the model to make this determination for itself as in LLM-PFC). For each problem, we selected the best out of five proposed plans (again based on a groundtruth evaluation). Although these decisions arguably gave ToT an advantage relative to LLM-PFC, we chose to evaluate ToT in
> > this way so as to give it the best possible chance of performing well in our task setting.'
> >
> > We found that ToT significantly underperformed on this task, solving only 25% of problems (as opposed to 74% in LLM-PFC), and proposing invalid actions 4% of the time (as opposed to 0% in LLM-PFC). This is likely due to the fact that each module is responsible for carrying out multiple functions, increasing the likelihood of errors. ToT especially tended to make errors in monitoring (error detection) and action proposal. This emphasizes the benefit of more extensively factorizing the planning procedure as in LLM-PFC. These results are now presented in Figure 3.
> >
> > [4] Yao, S., Yu, D., Zhao, J., Shafran, I., Griffiths, T. L., Cao, Y., & Narasimhan, K. (2023). Tree of thoughts: Deliberate problem solving with large language models. arXiv preprint arXiv:2305.10601.
> >
> > We would like to thank the reviewer once again for their helpful suggestions and comments. Please let us know if there are any remaining issues to address.

---

> > > ### Comment · Reviewer_t84f · 2023-11-23
> > > **Thanks for the response**
> > >
> > > Many thanks for your responses. I have raised my score, though I still maintain the same reservations as my original comment.

---

> > > > ### Author Response · Authors · 2023-11-23
> > > > **Response to reviewer t84f pt.4**
> > > >
> > > > Thank you very much for the response. We have now added the following additional result to the revised paper.
> > > >
> > > > ## Additional baseline
> > > >
> > > > We have added another baseline, multi-agent debate [5]. In this baseline, a solution is generated through debate between multiple LLM instances (each an instance of the GPT-4 ICL baseline), but these instances are not specialized to perform specific functions. Thus, the baseline tests for the extent to which LLM-PFC's planning performance depends specifically on the specialized nature of the modules vs. interaction between multiple instances in general. This baseline also underperformed on tower of hanoi, solving only 25% of problems correctly (vs. 74% for LLM-PFC). We have added this result to Figure 3.
> > > >
> > > > [5] Du, Y., Li, S., Torralba, A., Tenenbaum, J. B., & Mordatch, I. (2023). Improving Factuality and Reasoning in Language Models through Multiagent Debate. arXiv preprint arXiv:2305.14325.

---

### Meta-Review · Area_Chair_ZHrK · 2023-12-05

**Metareview:**

This paper presents a model for planning that combines multiple LLMs prompted to perform different tasks. The tasks that each LLM receives are inspired by the functional architecture of the human prefrontal cortex (PFC): a task decomposer and an orchestrator (akin to anterior PFC), an actor (akin to Dorsolateral PFC), a monitor for detecting conflicts (akin to anterior cingulate cortex), and a predictor and evaluator (akin to orbitofrontal cortex). The authors show that this model is better than a variety of baseline models at some tasks that involve planning, including graph traversal, the Tower of Hanoi, and a logistics task.

The reviewers were fairly critical of the initial submission, with consistent concerns about limited empirical evaluations and baseline comparisons. As such, the reviewers were not convinced that the model was convincing as an improvement on other techniques for planning. There were also concerns raised about computational costs of this approach.

The authors provided a robust set of rebuttals that included new tasks and new baseline models. However, though some scores were raised, the reviewers were not ultimately convinced by these revisions. In post-rebuttal discussion amongst the reviewers there was a general agreement that the additions helped the paper improve, but getting enough evaluations and additional baselines to really test the model capabilities fully would require more work still. As well, there were the lingering concerns about computational cost. In discussion, 3/4 reviewers recommended rejection, and the fourth was ambivalent.

This was a challenging decision as an AC, given that the authors did clearly work hard to address the reviewer concerns and there were no major technical concerns. But, it is true that the initial submission contained relatively few tests and baselines, such that the authors were fighting an uphill battle to get enough in the paper to convince the reviewers that this is a real advance. Given the post-rebuttal discussion, with a clear majority of reviewers advocating for rejection, I did not feel I could accept this paper. However, I encourage the authors to revise and submit elsewhere, given that the major concerns were a question of adding to the empirical evaluations, which can surely be addressed with more time.

**Justification For Why Not Higher Score:**

Per my points above, I don't feel that this paper is enough of an advance to warrant accepting it when a majority of reviewers recommend rejecting it, and the remaining one is ambivalent. This was a tough case though, and I think it could also be accepted post revisions.

**Justification For Why Not Lower Score:**

N/A

---

### Decision · Program_Chairs · 2024-01-16

Reject